# In-depth organic mass cytometry reveals differential contents of 3-hydroxybutanoic acid at the single-cell level

Shaojie Qin [1], Yi Zhang [1], Mingying Shi[1], Daiyu Miao [1], Jiansen Lu [2], Lu Wen [2] & Yu Bai [1] ✉

Comprehensive single-cell metabolic profiling is critical for revealing phenotypic heterogeneity and elucidating the molecular mechanisms underlying biological processes. However, single-cell metabolomics remains challenging because of the limited metabolite coverage and inability to discriminate isomers. Herein, we establish a single-cell metabolomics platform for in-depth organic mass cytometry. Extended single-cell analysis time guarantees sufficient MS/MS acquisition for metabolite identification and the isomers discrimination while online sampling ensures the high-throughput of the method. The largest number of identified metabolites (approximately 600) are achieved in single cells and fine subtyping of MCF-7 cells is first demonstrated by an investigation on the differential levels of 3-hydroxybutanoic acid among clusters. Single-cell transcriptome analysis reveals differences in the expression of 3-hydroxybutanoic acid downstream antioxidative stress genes, such as metallothionein 2 (MT2A), while a fluorescence-activated cell sorting assay confirms the positive relationship between 3-hydroxybutanoic acid and target proteins; these results suggest that the heterogeneity of 3-hydroxybutanoic acid provides cancer cells with different ability to resist surrounding oxidative stress. Our method paves the way for deep single-cell metabolome profiling and investigations on the physiological and pathological processes that occur during cancer.

Cellular heterogeneity plays an important role in many key biological processes, such as tumorigenesis, aging, immunity and development[1,2]. Research on single-cell omics provides global insights into heterogeneous molecular networks and phenotypic differences by combining multidimensional parameters in individual live cells[3–7]. Compared with the genome, transcriptome, and proteome, the single-cell metabolome focuses on endogenous metabolites and related biochemical reactions in individual cells, providing an ideal multiparameter approach to systematically understand the physiological and functional status of cells[8–15].

However, the small size of cell, complex composition, limited amount of metabolites without amplification capability make single-cell metabolomics more challenging. Mass spectrometry (MS) has emerged in single-cell metabolomics owing to its high sensitivity, high throughput, rapid response and powerful ability to identify structures[16,17]. In contrast to off-line discontinuous single-cell analysis, which involves individually extracting cellular contents through micromatulation[18], online single-cell metabolomics analysis involves continuously analyzing metabolites within individual cells in a monodispersed state using microfluidic devices. Typically, mass

[1]Beijing National Laboratory for Molecular Sciences, Key Laboratory of Bioorganic Chemistry and Molecular Engineering of Ministry of Education, College of Chemistry and Molecular Engineering, Peking University, Beijing, China. [2]Biomedical Pioneering Innovative Center, School of Life Sciences, Peking University, Beijing, China. ✉e-mail: yu.bai@pku.edu.cn

cytometry provides high throughput analysis of single cells through combining the continuous flow injection with MS that can provide accurate mass to charge ratios (m/z). Recently, the use of commercial cytometry by time of flight (CyTOF) in cell typing and immunity studies based on inductively coupled plasma–mass spectrometry (ICP-MS) has increased rapidly[19,20], which obtained protein information through detecting labeled metal ions without information on endogenous metabolites in the cell. In contrast, electrospray ionization (ESI)-based mass cytometry, such as CyESI-MS[21–23] and organic mass cytometry[24], enables researchers to continuously and efficiently acquire metabolome data in single cells. Nonetheless, MS/MS information of metabolites can be deficient due to limited analysis time or insufficient release of cellular content, which hampers metabolite identification and in-depth single-cell metabolomic analysis. Due to the above factors, the discrimination of isomers, which contributes to precise metabolite identification and distinctive functional elucidation, is more challenging than that of other metabolites[25,26]. Although the C=C bond positions in lipid species[27,28] and D-lactate[29] were analyzed with the help of derivatization and enzymatic reactions, general approaches to analyze isomers at the single-cell level are still lacking. Furthermore, it remains challenging to intensively and accurately profile metabolites without losing cell throughput.

Herein, we develop an in-depth organic mass cytometry platform (ID-organic cytoMS), which involves continuous cell sampling, efficient online cell lysis, and non-contact ESI-MS followed by metabolomic analysis (Fig. 1). Cell mono-dispersion is realized via Dean flow[30] in an ordering capillary, which is subsequently enclosed by a coaxial sheath-liquid capillary through a T-junction. In our method, online cell lysis is achieved when cells meet the sheath liquid and go through ultrasonication in an ice bath. The average analysis time is 25 s for each single cell in continuous flow mode, which facilitates the acquisition of high-resolution MS and MS/MS data. Comprehensive metabolic profiling and acquisition of MS/MS spectra in a single cell are successfully achieved in an online manner. A total of 224 and 348 metabolites are identified in single MCF-7 cells using MS/MS information in negative and positive mode, respectively. Sufficient MS/MS data provides structural information and the relative abundance of isomers, providing a demonstration of MCF-7 cell subtyping with the extra dimension of 3-hydroxybutanoic acid (BHB). Single-cell transcriptome analysis and fluorescence-activated cell sorting (FACS) assays reveal positive relationship between BHB and its downstream target proteins involved in antioxidative stress. This work proposes and validates an approach for high-throughput and in-depth single-cell metabolomics, which paves the way for comprehensive clarifying metabolic heterogeneity in cells and interpreting complicated biological processes at the single-cell level.

## Results

### Configuration and performance of in-depth organic mass cytometry

The short MS data acquisition time of each single cell in CyESI or organic mass cytometry limits the ability to accurately identify and interpret metabolite information. Currently, to identify metabolites in single cells, researchers generally match accurate masses created from bulk cells; however, the metabolites are identified ambiguously. Therefore, efficient online cell lysis and sufficient analysis time are particularly important for MS/MS data acquisition and abundant metabolite identification. Based on the organic mass cytometry platform constructed by our group[24], ID-organic cytoMS was further developed with an efficient online lysis system and high-resolution non-contact ESI-MS (Fig. 1 and Supplementary Fig. 1a). In this system, MCF-7 cells were suspended in 140 mM ammonium formate, which helps cells maintain an intact cell state. Then, the cell suspension was pumped into the inner ordering capillary in which single-cell sorting was achieved via Dean flow. Low-energy ultrasonication was employed to realize online cell lysis after the cell solution mixed with sheath fluid. In addition, ice bath conditions were used to prevent metabolite degeneration caused by the heating effect from continuous ultrasonication. In order to enhance the ionization efficiency of metabolites and improve the cell lysis performance, a sheath liquid of methanol with 1% ammonia solution or 1‰ formic acid was added through the T-junction and outer capillary. Segmented cell lysates originating from single cells were ionized using homemade direct current (DC) non-contact ESI (Supplementary Fig. 1b) and analyzed by a high-resolution Orbitrap mass spectrometer. Thus, compared with that of previous work, a longer MS data acquisition window of single cells was successfully achieved[21–24]. A coaxial sheath-liquid capillary with a tapered emitter is directly utilized for electrospray to prevent cell debris from clogging the system and to continuously analyze the cells. To verify that the single cells were isolated, the cells were preliminarily stained with the nuclear dye Hoechst 33342 and the membrane dye DIO before resuspension in ammonium formate. Supplementary Fig. 2 shows that cells were isolated individually within capillaries. After sonication, obvious cell shrinkage occurred and the fluorescence signal of the cell nucleus and membrane almost completely disappeared, which confirmed that the cell structure was destroyed (Supplementary Fig. 3). Methanol from the sheath liquid helps extract the cellular content and

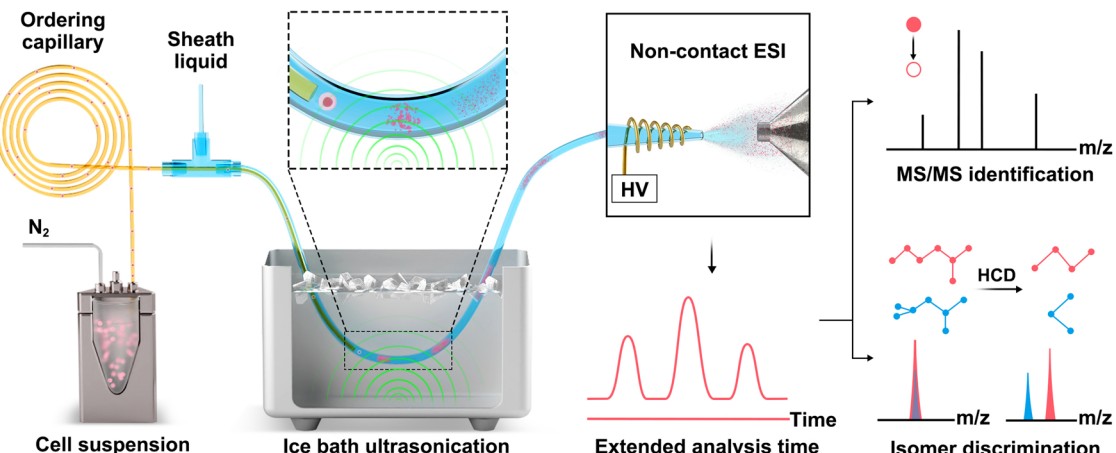

**Fig. 1 | The workflow of ID-organic cytoMS.** The cell suspension was pumped into an ordering capillary to achieve single-cell monodispersion, and intact single cells were lysed online after they left the ordering capillary and mixed with sheath liquid via ice-bath ultrasonication. Segmented lysate from single cells was subsequently analyzed by non-contact ESI-MS. Extended analysis time ensures that abundant MS/MS data are obtained and isomer discrimination occurs at the single-cell level.

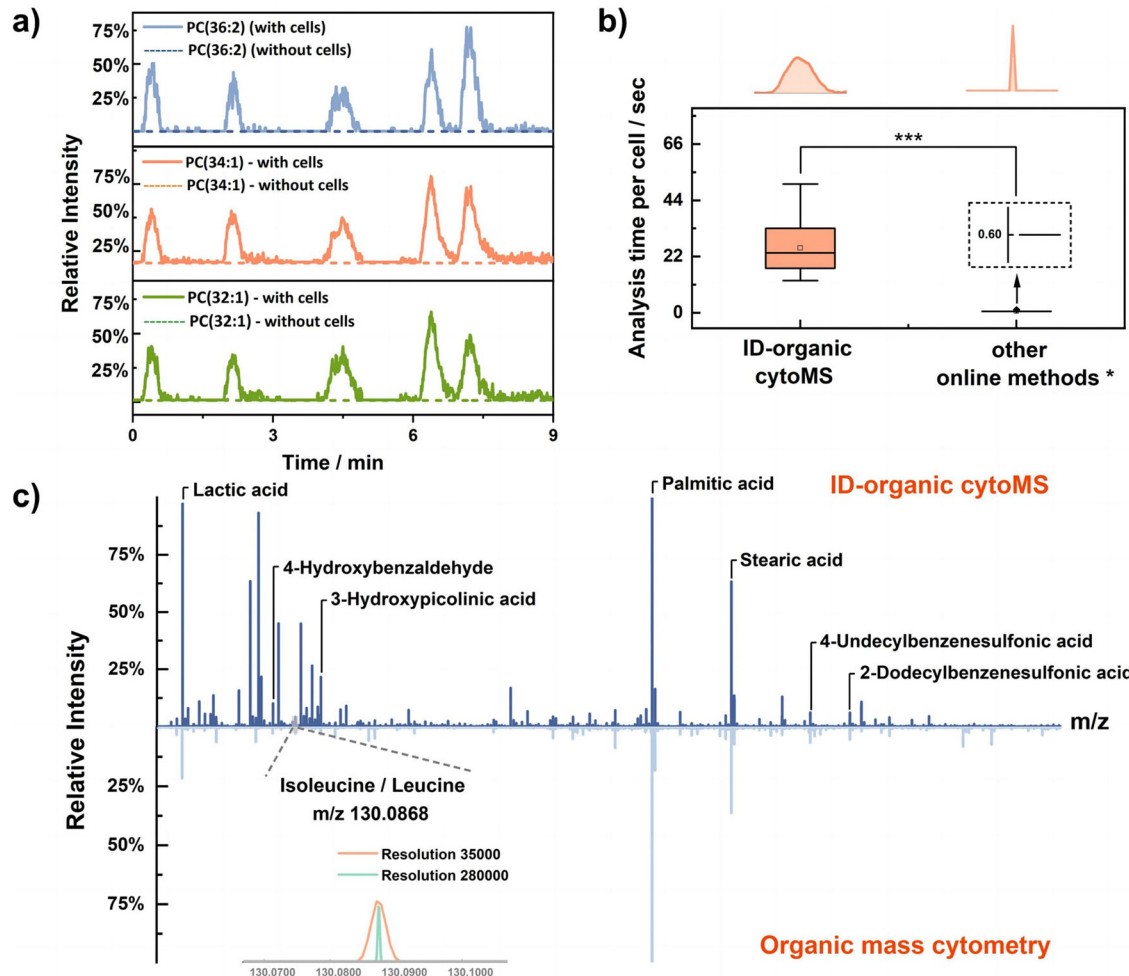

**Fig. 2 | Validation of ID-organic cytoMS. a** EICs of the endogenous metabolites PC (34:1), PC (32:1) and PC (36:2) with or without cells. **b** Analysis-time extension performance of ID-organic cytoMS and other online methods (Boxplot shows median, 0.25 and 0.75 quantile, and whiskers extend to points within 1.5 inter-quartile range of lower and upper quartile. $n = 100$ cells, Wilcoxon test, two-sided,

$P = 0$, *** denoted $P < 0.001$, *: all other online mass cytometry using ESI-MS). **c** Representative single-cell MS spectrum of ID-organic cytoMS and organic mass cytometry (negative ion mode, $m/z$: 80-400). The MS peaks at $m/z$ 130.0868 (leu-cine/isoleucine) with mass resolutions of 35,000 and 280,000, respectively. Source data are provided as a Source Data file for **a** and **c**.

ensures that the ionization spray remains stable during the analysis. Online single-cell lysis and metabolite diffusion caused by sonication results in the extension of the single-cell MS detection window. The MS data showed that normally distributed signals appeared in the extracted ion chromatograms (EICs) of cell-specific lipids, such as PC (34:1), PC (32:1) and PC (36:2) (Fig. 2a). In contrast, no signal emerged in the corresponding EICs in the ammonium formate solution, indicating that normally distributed pulses were derived from cell events. The number of cells flowing through capillaries that was determined by microscopy was compared with the pulse number collected by ID-organic cytoMS within 20 min (Supplementary Fig. 4), and nearly identical results were obtained, confirming the one-to-one corre-spondence between pulse signals and individual cell events. Compared with previous organic mass cytometry methods, this method sig-nificantly improved the single-cell analysis time from the original 0.6–25 s, resulting in a more than 40-fold increase (Fig. 2b). So far, our method displayed the longest online MS detection window of single cells when compared with all literatures, and the extended analysis time enabled the acquisition of sufficient MS/MS data for metabolite identification.

With online lysis and the extension of analysis time, ID-organic cytoMS has the following advantages. In terms of MS peaks, ID-organic cytoMS detected 534 and 1285 characteristic MS peaks in negative and

positive mode, respectively (Fig. 2c and Supplementary Fig. 5). The dominant peaks from ID-organic cytoMS were assigned to fatty acid metabolism, glycolysis, the TCA cycle, etc. Improved metabolite cov-erage probably results from sufficient cell lysis caused by sonication. Moreover, the extended single-cell analysis time provides sufficient time to conduct high-resolution MS acquisition (Fig. 2c). Taking the MS peak at m/z 130.0868 (leucine/isoleucine) as an example, a mass resolution of 280,000 clearly offers narrower half-width of the MS peak than that obtained with a mass resolution of 35,000, which enhance the accuracy of metabolite identification. Taken together, the results indicated that the ID-organic cytoMS lays a foundation for future studies on advanced metabolic profiling and more accurately metabolites identification.

To achieve the best performance in the extended lysis of single cells, relevant parameters, including the cell suspension density (Supplementary Fig. 6) and sheath liquid flow velocity (Supplementary Fig. 7), were further optimized. Methanol supplemented with 1% ammonia solution or 1‰ formic acid was chosen as the sheath liquid based on the stability of spraying, cell rupture performance[31] and MS response. To achieve higher cell analysis throughput and generate an undisturbed single-cell signal, ~7000 cells/ml cell suspension and 10 µL/min of sheath liquid velocity were chosen for subsequent experiments. Other parameters, such as capillary diameter and

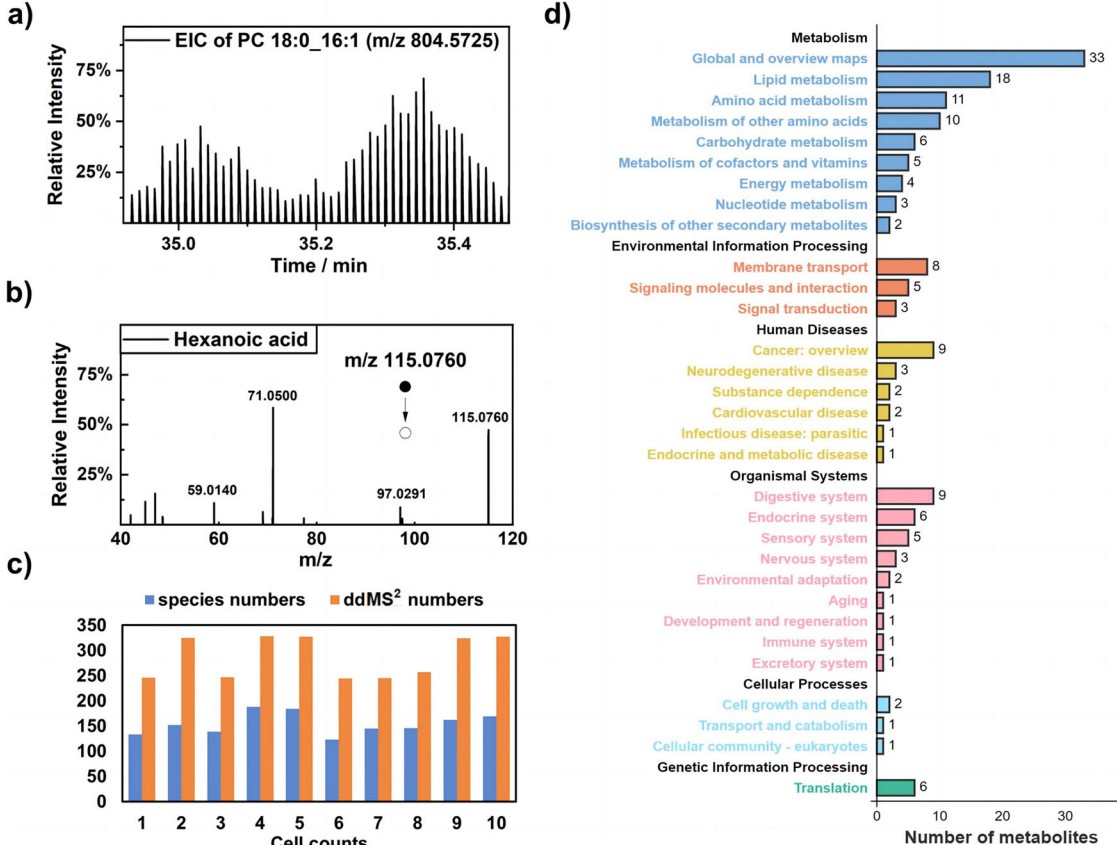

**Fig. 3 | Profiling of metabolites according to MS² at the single-cell level.**
**a** Typical EIC of PC (18:0_16:1) within two single cells with the acquisition mode of Full MS-ddMS² (Top7). Source data are provided as a Source Data file for **a**. **b** MS² spectrum of hexanoic acid in a single cell. **c** Numbers of species identified by using MS² and numbers of ddMS² achieved in 10 single cells. **d** KEGG pathway annotation results for the identified metabolites in 120 single cells.

sonication power, were optimized based on the literature[30,32,33]. In addition, ice-bath sonication was performed in a non-contact manner to prevent interference from external energy and the degradation of endogenous compounds. Taken together, the above results verified that our method enabled the continuous lysis of single cells assisted by low-temperature ice-bath ultrasonication. Our method offered a robust tool for conducting comprehensive and in-depth research on single-cell metabolomics.

### Deep profiling of single-cell metabolites using MS²

With its ability to perform extended analyses on single cells, ID-organic cytoMS enables MS² acquisition within single cells, which offers great possibilities for researchers to elucidate metabolite structures and identify metabolite species. We conducted consecutive single-cell metabolome data acquisition using TopNddMS² ($n = 7$) to establish an MS/MS database with high-resolution MS data. As shown in Fig. 3a, the discrete ion chromatogram mainly resulted from the intensity difference between the full MS spectrum and the following MS/MS spectrum, and Fig. 3b displays the acquired MS² spectrum of a representative metabolite of hexanoic acid in a single cell. With this process, an average of 300 ddMS² acquisitions and 150 species were identified from one MCF-7 cell (Fig. 3c). Overall, 224 and 348 metabolites were successfully identified in negative and positive mode, respectively, from single MCF-7 cells (Supplementary Dataset 1). Through our method, the largest number of metabolites were identified in single cells to date in an online manner. These metabolites principally include carboxylic acids, amino acids, fatty acids, lipids and their derivatives. Kyoto Encyclopedia of Genes and Genomes (KEGG) annotation revealed that these metabolites are involved mainly in basal energy metabolism, membrane transport and other cellular

processes (Fig. 3d and Supplementary Fig. 8). In summary, the ID-organic cytoMS platform provided sufficient time for MS² acquisition and was able to realize metabolite identification in single cells with high cell throughput. This approach offered great possibilities for researchers to speculate metabolite structures or even discriminate isomers and was a promising method for interpreting cell metabolic heterogeneity.

### Discrimination of isomers in single MCF-7 cells

Compared with metabolite identification, isomer discrimination is more challenging during single-cell analyses. With our method, which benefits from the extended analysis time with online cell lysis, structural isomers can be generally discriminated via sufficient MS² data. As shown in Fig. 4a–c, 3 pairs of isomers (PC (15:1_18:1) and PE (18:1_18:1) at $m/z$ 744.5543; 5-aminovaleric acid and valine at $m/z$ 118.0863; and 4-hydroxybutanoic acid and 3-hydroxybutanoic at $m/z$ 103.0401) were distinguished in single cells based on their unique MS/MS spectra. Specific diagnostic ion pairs ($m/z$ 184.0733 and 603.5352, $m/z$ 72.0808 and 101.0597, $m/z$ 57.0346 and 59.0138) were highlighted with different color. Their reference MS² spectra together with the proposed fragmentation positions were shown in Supplementary Fig. 9. For the relative quantification of isomers among single cells, the relative abundance of the diagnostic ion was used according to literature[27]. Figure 4d−f shows the relative abundance ratio of the isomer pairs among 60 single cells. Interestingly, BHB and GHB showed greater differences than those of the other two pairs, which were confined by a maximum of one magnitude. The MS² spectrum at $m/z$ 103.0401 also exhibited heterogeneity in 12 random cells (Supplementary Fig. 10). The MS² spectra of the BHB standard with different CID collision energies are shown in Supplementary Fig. 11, which confirmed the

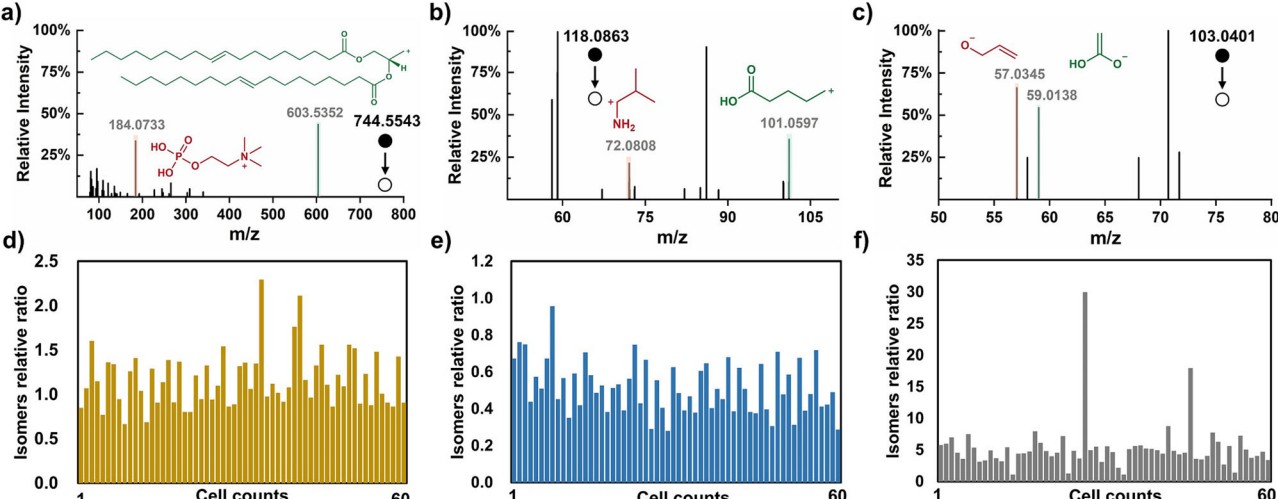

**Fig. 4 | Qualitative and quantitative analysis of isomers at single-cell level.**
Single-cell MS² spectra of ions at **a** *m/z* 744.5543 (PC (15:1_18:1) and PE (18:1_18:1)); **b** *m/z* 118.0863 (5-aminovaleric acid and valine); and **c** *m/z* 103.0401 (4-hydroxybutanoic acid and 3-hydroxybutanoic acid). (Characteristic fragment ions and corresponding MS peaks are highlighted in orange or green). The relative abundance of the above isomers among 60 single cells is shown in **d–f**. The relative isomer ratio was defined as the intensity ratio between the characteristic fragment ions. (*m/z* 184.0733 vs. *m/z* 603.5352; *m/z* 101.0597 vs. *m/z* 72.0808; *m/z* 59.0138 vs. *m/z* 57.0345). Source data are provided as a Source Data file for **a–f**.

specificity of the diagnostic ions. The above findings likely imply that the relevant BHB/GHB pathway in MCF-7 cells is significantly heterogenous, and the detailed mechanism needs to be further explored.

### Refined subtyping of MCF-7 cells using BHB/GHB isomer pairs

Before the subtyping potential of isomer pairs was further explored, the feasibility of ID-organic cytoMS in conventional cell typing was investigated. Metabolic profiling of 4 types of tumor cells, namely, MCF-7, MDA-MB-231, HeLa, and HepG2 tumor cells, was performed. As shown in Fig. 5a, four types of tumor cells were successfully distinguished by the abundance of metabolites without isomers by t-distributed stochastic neighbor embedding (t-SNE). A heatmap of the single-cell metabolite profiles of the four tumor cell types revealed distinctive metabolic heterogeneity among the 4 kinds of cell types (Supplementary Fig. 12). Notably, no obvious clustering was observed within MCF-7 cells based on the above metabolite data from single cells (Fig. 5b). Conversely, after the BHB/GHB isomer information was combined, 3 clusters of cells were distinguished (Fig. 5b, Supplementary Fig. 13). Relative ratio of BHB/GHB isomers from 3 clusters was significantly different among clusters (Fig. 5c). The other two isomers, however, cannot further distinguish between subtypes of MCF-7 cells (Supplementary Figs. 14 and 15). A volcano plot of metabolites among clusters 1, 2 and 3 (Supplementary Fig. 16) revealed that no significant differences in metabolite abundance were observed. Notably, MCF-7 cells are derived from breast tumors and are dominated by the luminal A subtype[34]. This is a demonstration of MCF-7 cell subtyping by isomers in single cells, and our method provides insights on tumor heterogeneity and cell types subdividing for precision therapy of breast tumors.

### Multi-omics data interpretation of BHB in MCF-7 cells

Next, 10×Genomic single-cell transcriptome sequencing was performed with 10373 single MCF-7 cells to further verify the possible mechanism underlying the differential abundance of BHB/GHB among MCF-7 cells. The accuracy of the sequencing results was initially validated by verifying the expression of specific proteins in MCF-7 cells (Supplementary Fig 18). As shown in Fig. 6a, the sequencing data revealed that the MCF-7 cells exhibited three distinct clusters, consistent with the single-cell metabolomics subtyping results. To further elucidate whether this cell subtype differentiation is attributed to BHB

or GHB, we conducted unbiased differential expression analysis (Supplementary Fig 19) and focused on relevant proteins, including enzymes involved in BHB/GHB generation, consumption and transport, as well as downstream target proteins. First, GHB metabolism-related enzymes, including *PON3* (Lactonase 3) and *AKR7A2* (aldo-keto reductase family 7 member A2) for GHB production and *ALDH5A1* (aldehyde dehydrogenase 5 family member A1) for the GHB dehydrogenation process, were analyzed. No significant difference in the expression of the above genes were observed among the MCF-7 cells (Supplementary Fig. 20). In addition, the Western blot results confirmed these findings by the protein expression level (Supplementary Fig. 21). The above results suggested that the cluster discrimination was probably attributed to the differential abundance of BHB rather than GHB.

BHB, the dominant ketone body in the blood or brain, plays an essential role in supplying energy when glucose cannot satisfy the body's energetic needs[35] by accelerating the TCA cycle[36]. Recent research has shown that BHB is produced in tumor cells[37], adipocytes[38] and cancer-associated fibroblasts[39]. BHB is increasingly regarded as a cellular signaling metabolite that links the cellular metabolic status and posttranslational modifications (PTMs) of histone proteins as well as further transcriptional activity[35,40] and is associated with the expression of genes related to aging, oxidative stress resistance and neurodegenerative diseases[41,42]. Shimazu et al.[40] reported that BHB, which acts as an endogenous histone deacetylase (HDAC) inhibitor, promotes the transcription of the oxidative stress resistance factor *MT2A*. Differential gene expression analysis revealed differential *MT2A* expression among the clusters (Fig. 6b, Supplementary Fig. 19 and Supplementary Fig. 22). In addition, heterogeneous nuclear ribonucleoprotein A1 (hnRNP A1), which plays a crucial role in alternative mRNA splicing and transcriptional regulation[43], was reported to bind directly with BHB to prevent vascular[44] and cartilage[45] from oxidative stress. Our results also revealed differential expression of *hnRNP-A1* among the clusters (Fig. 6b and Supplementary Figs. 19, 22). The distinct changes in the expression of the above proteins likely corresponded to differences in the BHB content. Considering the significant differential expression and overall high expression proportion (Supplementary Fig. 23), we selected MT2A and hnRNP A1 as sorting markers to further investigate the relationship between BHB abundance and target protein expression. MT2A and hnRNP A1 antibodies

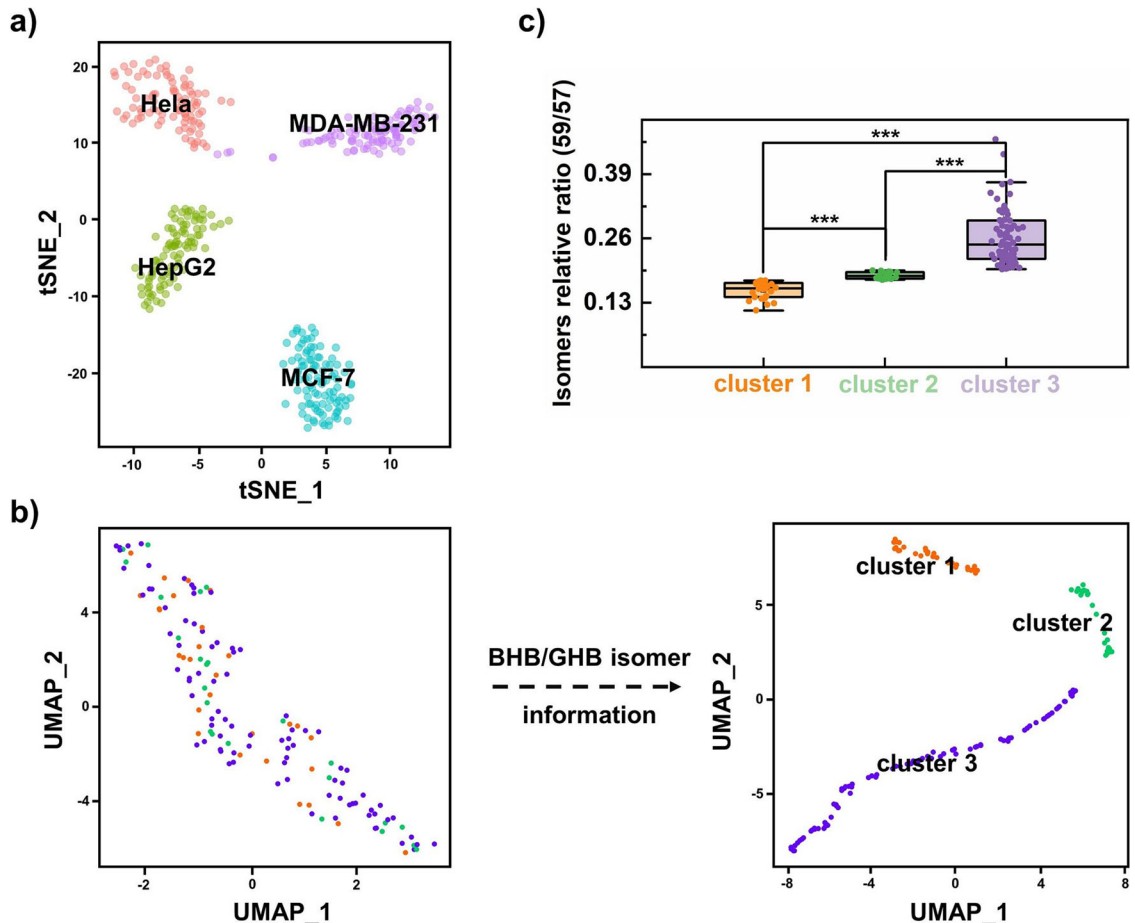

**Fig. 5 | Clustering of MCF-7 cells with BHB/GHB isomer information. a** t-SNE visualization of 4 types of tumor cells. **b** Uniform manifold approximation and projection (UMAP) visualization of MCF-7 cells with and without BHB/GHB isomers (*n* = 138 cells). **c** Relative abundance of BHB/GHB isomers (MS peak intensity at *m/z* 59.0138 vs. *m/z* 57.0345) among 3 clusters (*n* = 138 cells in total). The boxplot shows the median and 0.25 and 0.75 quantiles, and the whiskers extend to points within the 1.5 interquartile range of the lower and upper quartiles. (Kruskal–Wallis test, *P* = 4.3E-23, *** denotes *P* < 0.001). Source data are provided as a Source Data file for **c**.

conjugated with fluorophores were utilized for indirect FACS, and then, the collected cells with different MT2A/hnRNP A1 expression were analyzed by ID-organic cytoMS. As shown in Fig. 6c, d, the cell population with greater expression of MT2A/hnRNP A1 exhibited greater relative abundance of BHB, which verified the linear positive relationship between BHB content and target protein expression (Pearson's *r* = 0.99). In addition, significant differential expression was observed for other BHB target genes related to antioxidative stress and aging progression, including *SOD*[40,46,47], *SULT2B1*[48,49], and *GATA3*[50] (Supplementary Figs. 19 and 22). These genes are involved in oxidative stress defense, tumor suppression, cell senescence etc., which implies multiple roles of BHB in cancer cells.

In addition to acting as an epigenetic regulator, BHB has been reported as an energy source in normal cells that provides substrates (acetyl-CoA) for the subsequent TCA cycle and fatty acid metabolism[35,41]. However, single-cell metabolomic results show that abundance of TCA cycle (Supplementary Fig. 24a) and fatty acids metabolism (Supplementary Fig. 24b) related metabolites were not correlated with the content of BHB among the 3 clusters. It was proposed that cancer cells cannot utilize BHB as an alternative energy source owing to metabolic inflexibility caused by mitochondrial abnormalities[38,51], while our work demonstrates and experimentally evidences it at the single-cell metabolomic level. In summary, differences content of BHB occurred in MCF-7 cells, which affects target proteins related to signal transduction pathways rather than energy supply.

Furthermore, KEGG enrichment analysis revealed that the oxidative phosphorylation pathway was significantly enriched, as well as various pathways associated with neurodegenerative diseases such as Parkinson's disease and Alzheimer's disease, which aligns closely with the physiological functions of BHB (Supplementary Fig. 25). Several differentially expressed genes among clusters involved in transcription regulation, stress response, and neurodevelopmental disorders, including *KCNQ1OT1*[52] and *NEAT1*[53,54], may also play dispensable roles in BHB-mediated signal transduction (Supplementary Dataset 2). For instance, *NEAT1* was found to be upregulated in HDAC inhibition-tolerant cells[55], which may be related to BHB-mediated antioxidation. Our work is a pioneer in the study of BHB and cancer at the level of single-cell metabolomics and transcriptomic analysis, which opens up new insight and requires further in-depth investigations.

## Discussion
In this study, we designed the ID-organic cytoMS technique by proposing an online cell lysis strategy and integrating with cell sorting and high-resolution ESI-MS. Isotonic ammonium formate buffer was utilized to maintain an intact cell state. Notably, low-energy ice-bath ultrasonication prevents metabolite degeneration caused by heating effects from continuous ultrasonication. Custom-made DC non-contact ESI and high-resolution Orbitrap mass spectrometer were used for single-cell metabolomic data acquisition. The average 25 s analysis time for each single cell, 40 times greater than that of other

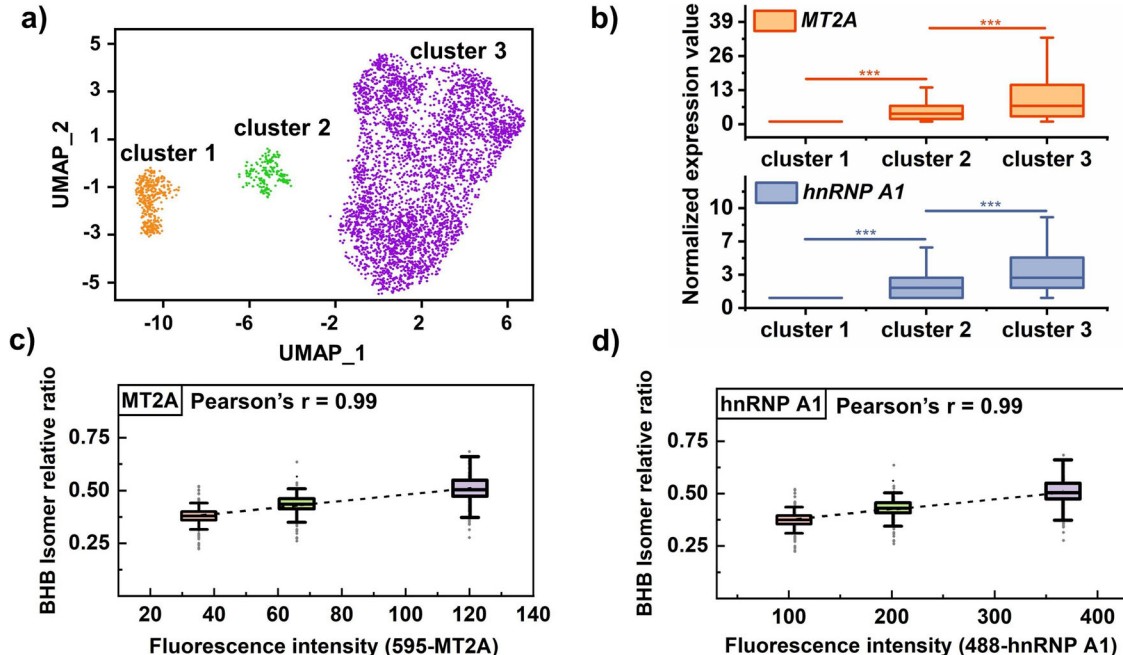

**Fig. 6 | 10x Genomic single-cell transcriptome sequencing verification. a** UMAP visualization of the single-cell sequencing results for MCF-7 cells. The three cell clusters are distinguished by different colors. **b** Box plot of the normalized expression levels of *MT2A* (upper, $P = 1.18\text{E-}123$) and *hnRNP A1* (lower, $P = 6.25\text{E-}117$) among the clusters ($n = 10373$ cells in total). (Kruskal–Wallis test, *** denoted $P < 0.001$) **c**. Box plot of BHB relative abundance among cell populations with different expression levels of MT2A. ($n = 216$ cells) **d**. Box plot of the relative abundance of BHB among cell populations with different hnRNP A1 expression levels ($n = 216$ cells). All of the above boxplots show the median and the 0.25 and 0.75 quantiles, and the whiskers extend to points within the 1.5 interquartile range of the lower and upper quartiles. Source data are provided as a Source Data file for **b**–**d**.

online modes, was achieved with the continuous flow mode. Overall, 224 and 348 metabolites were the most abundant metabolites identified in single MCF-7 cells using MS/MS in negative and positive mode, respectively. Sufficient MS/MS data provides the possibility for isomer identification and their relative abundance determination, which demonstrates that MCF-7 cell subtyping can be performed using the additional information of BHB abundance. 10×Genomics single-cell transcriptome sequencing revealed that MCF-7 cells exhibited three distinct clusters, consistent with the single-cell metabolomics sub-typing results. Unbiased analysis of differentially expressed genes revealed significant difference of anti-oxidative stress relevant genes *MT2A* and *hnRNP A1*. The expression levels of MT2A and hnRNP A1, genes encoding downstream target proteins associated with the anti-oxidant stress response, exhibited a positive relationship with the abundance of BHB. Contents of metabolites of TCA cycle and fatty acids synthesis were not correlated with that of BHB in MCF-7 cells, which suggested that cancer cells cannot utilize BHB as an alternative energy source. Overall considering the antioxidant stress and anti-aging functions of BHB target proteins including SOD, GATA3 and SULT2B1, the differences in BHB content may imply cancer cells with various ability to resist surrounding stress.

In summary, we established a single-cell online extended lysis strategy coupled with high-resolution MS analysis. Through the incorporation of low-energy sonication in sheath liquid, isolated single cells could be lysed online for more sufficient content release, which involves several advantages, including high metabolite coverage, high cell throughput and adequate single-cell analysis time. Through the extended analysis time, abundant MS/MS data were collected at the single-cell level, which facilitate the accurate identification of meta-bolites and the precise discrimination of isomer structures. Notably, MCF-7 cells were further subtyped with the BHB isomer information. The results from both single-cell transcriptome sequencing and FACS confirmed these findings. Considering various physiological functions of BHB, BHB pathway-targeted inhibitors may broaden the scope of

drug candidates for current breast cancer therapies. Our method provides a powerful platform for performing single-cell metabolome analysis in a comprehensive and accurate manner for fundamental cell biology and cancer therapy research.

## Methods

### Chemicals and materials

Cells and cell culture: MCF-7 cells were purchased from Union Cell Resource Center (Beijing, China), and MDA-MB-231, HepG2 and HeLa cells were kind gifts from Xinxiang Zhang Laboratory (Peking University). Dulbecco's modified Eagle's medium (DMEM)/high glucose (+0.584 g/L L-glutamine, +4.5 g/L glucose, +3.7 g/L sodium bicarbonate, +0.110 g/L sodium pyruvate, +80 U/mL penicillin, +0.08 mg/mL streptomycin) was purchased from Thermo Fisher Scientific Life Technologies (USA). McCoy's 5 A medium (+0.219 g/L glutamine, +80 U/mL penicillin, +0.08 mg/mL streptomycin) was purchased from Dalian MeilunBio. Co., Ltd. (Dalian, China). RPMI-1640 medium (+0.3 g/L glutamine, +80 U/mL penicillin, +0.08 mg/mL streptomycin) was purchased from Nanjing KeyGen Biotech Co., Ltd. (Nanjing, China). Trypsin-EDTA digestion solution (0.25%) was purchased from Solarbio Biotech. Co., Ltd., Beijing, China). Anti-hnRNP A1 and anti-MT2A antibodies were purchased from Abcam (USA). Phosphate-buffered saline (PBS 1×) (- Calcium, - Magnesium) and 4% paraformaldehyde solution were purchased from HyClone (Logan, UT, USA). Certified fetal bovine serum (FBS) was purchased from BI (Kibbutz Beit-Haemek, Israel). Methanol (HPLC grade) and ammonia solution (HPLC grade) were purchased from Fisher Scientific (USA). Purified water was obtained from Hangzhou Wahaha Group (Hangzhou, China). HPLC-grade formic acid was purchased from Aladdin (Beijing, China). Ammonium formate (HPLC grade) was purchased from Macklin (Beijing, China). 3-Hydroxybutyric acid was purchased from Aikang Biomedical R&D. Co., Ltd., Jiangsu, China). DIO was purchased from Jinming Biotechnology Co., Ltd. (Beijing, China), and Hoechst 33342 staining solution was purchased from Beyotime Biotechnology Co., Ltd.

(Shanghai, China). Ordering silica capillary (150 μm i.d. and 360 μm o.d.) were obtained from Ruifeng Chromatographic Devices Co., Ltd. (Hebei, China). Electrospray emitters were made of fused silica capillaries (360 μm i.d. and 150 μm o.d.) obtained from Ruifeng Chromatographic Devices Co., Ltd. (Hebei, China). An ultrasonic cleaner (KM410C) was purchased from Guangzhou Kemeng Clean Technology Co., Ltd. (Guangzhou, China), and the ultrasonic power and frequency were set to the default values. The pressure injection cell was purchased from Next Advance, Inc. (USA).

All antibodies were commercially purchased. Anti-Metallothionein antibody (Mouse monoclonal, 1/1000 dilution used, ab12228, UC1MT, GR3415395-4), Recombinant Anti-hnRNP A1 antibody (Rabbit monoclonal, 1/1000 dilution used, ab177152, EPR12768, GR139101-6), Recombinant ALDH5A1/SSADH antibody (Rabbit monoclonal, 1/10000 dilution used, ab129017, EPR7794, GR3440323-7), anti AKR7A2 antibody (Rabbit polyclonal, 1/500 dilution used, ab97458, GR73247-6) and anti PON3 antibody (Rabbit monoclonal, 1/1000 dilution used, ab109258, EPR2903(2), 1068361-1) were purchased from abcam. HRP-conjugated GAPDH Monoclonal antibody (Mouse monoclonal, 1/3000 dilution used, HRP-60004, 1E6D9, 21005148) were purchased from proteintech. Secondary antibody (green) Alexa Fluor® 488 donkey anti-rabbit IgG (H + L) (Donkey polyclonal, used at a 1/3000 dilution, invitrogen, R37118, 2376850) and Secondary antibody (red) Alexa Fluor® 594 donkey anti-mouse IgG (H + L) (Donkey polyclonal, used at a 1/3000 dilution, invitrogen, R37115, 2474956) were purchased from Thermo Scientific.

Fluorescence and bright-field images were acquired using a Nikon Eclipse Ti inverted fluorescence microscope system (Nikon, Japan). Fluorescence-based flow cytometric assays were performed using an Astrios EQ flow cytometer (BeckMan Coulter, USA).

## Cell culture and harvest

MDA-MB-231, HepG2 and HeLa cells were cultured in DMEM/high glucose medium supplemented with 10% FBS, 80 U/mL penicillin, and 80 μg/mL streptomycin in an incubator (5% $CO_2$, 37 °C). MCF-7 cells were cultured in RPMI-1640 medium supplemented with 10% FBS, 80 U/mL penicillin, and 80 μg/mL streptomycin in an incubator (5% $CO_2$, 37 °C). At the logarithmic growth phase, the medium was removed, and the cells were rinsed once with PBS and then dissociated with trypsin solution within 2–5 min. The supernatant was removed by centrifugation (300 g, 5 min), after which the cells were suspended in PBS for subsequent use. Before MS analysis, the PBS was replaced with 140 mM ammonium formate through centrifugation (300 g, 5 min). The cell number was determined using a cell counter, followed by dilution with ammonium formate to 7000 cells/ml. The cell suspension was placed on ice for later use.

## Cell staining and observation

DIO was diluted to a final concentration of 5 μM with Hoechst 33342 working solution to prepare the staining buffer. After the cells reached the logarithmic growth phase, the medium was removed, the cells were washed with PBS. Then, the cells were incubated with staining buffer at 37 °C for 30 min, the buffer was removed after incubation, and the cells were subjected to two rounds of centrifugal washing (300 g, 4 min) with PBS. Finally, the cells were resuspended in ammonium formate solution for subsequent fluorescence microscopy. Fluorescence excitation wavelengths of 405 and 488 nm were used to obtain fluorescence images of the nuclei and membrane, respectively.

## Online cell lysis and MS analysis

The mass cytometric analysis of the cells was performed as follows (Supplementary Fig. 1): The prepared cell solution (~7 × 10³/ml) was injected by a pressure injection cell with an $N_2$ pressure of 20 psi, and the cells were monodispersed by an ordering capillary twinned into 5

loops with diameters of 1 cm. Magnetic stirring was performed to maintain a homogeneous state of the cell suspension. The ordering capillary used was a silica capillary 40 μm i.d. and 100 μm o.d. Sheath liquid (methanol supplemented with 1% ammonia solution or 1‰ formic acid) was incorporated through the T-junction, and the ordering capillary was enclosed within the coaxial sheath liquid capillary at 150 μm i.d. and 360 μm o.d. The sheath liquid velocity was set to 10 μl/min. An approximately 5 cm sheath-liquid capillary was buried in an ice-bath ultrasonication cleaner. Approximately 5 cm of copper wire was wrapped around the emitter. MS analysis was performed in positive and negative mode. The flow system was washed after each test or between two tests using 140 mM ammonium formate solution with a $N_2$ pressure of 20 psi for 5 min.

Electrospray emitters were created directly using a sheath liquid capillary with fused silica capillaries (150 μm i.d. and 360 μm o.d.). The tip of the capillary was first disposed on fire and then washed with ethanol to peel off the coating of the capillary. Then, the capillary was pulled by a P-2000 (Sutter Instrument) to make the emitter with a tip of approximately 50 μm. The parameters of P-2000 were as follows: HEAT = 400, FIL = 5, VEL = 28, DEL = 125, and PUL = 80. The spray electric field was provided by an MS built-in DC electrical power supply system, and electric contact was realized through connection of the copper wire surrounding the emitters with a nanospray ion source (Thermo Fisher Scientific, CA, USA). All the mass cytometric experiments were conducted on an Orbitrap MS (Orbitrap Exploris 480, Thermo Fisher Scientific, CA, USA). The parameters of the full-scan MS mode were set as follows: mode, negative; scan range, $m/z$ 80-1200; resolution, 35000; microscans, 1; AGC target, 1e6; maximum injection time, 50 ms; sheath gas flow rate, 0; aux gas flow rate, 0; sweep gas flow rate, 0; spray voltage, 2.5 kV; capillary temperature, 320 °C; and tube lens voltage, 60. For metabolite identification at the single-cell level, the full MS-ddMS2 acquisition mode was used. For the full MS process, the parameters were set as follows: $m/z$ 80-1200, resolution: 35000, microscans: 1, AGC target: 3e6, and maximum injection time: 100 ms. For the ddMS2 process, the parameters were set as follows: resolution: 17500, AGC target: 1e5, maximum IT: 50 ms, loop count/top N: 7, isolation window: 0.4 $m/z$, NCE: 30,40,50, minimum AGC target: 8e3, and excluding isotopes: on, dynamic exclusion: 20 s. For 3-hydroxybutyric acid quantification in single cells, an inclusion list of $m/z$ 103.0400 was added to the above settings. Xcalibur software (version 4.0) was used for the control of the MS system, data collection and subsequent analysis.

## Single-cell data analysis and metabolite identification

Single-cell mass spectra were extracted from each normalized distribution peak (peaks >5 times the signal-to-noise ratio, pulse time >10 s in EICs) in the ion chromatograms, and the mass spectra with the highest total ion current were selected to represent single cells. The intensities of the identified metabolites in each mass spectrum of a single cell were assembled into a data matrix. The MS intensities of metabolites in each mass spectrum were normalized by the total ion current (TIC). Characteristic MS peaks were defined as new MS peaks that appeared in the sample spectra compared with those in the control group spectra. The metabolites were initially identified with the help of Compound Discoverer 3.3, which is configured with the mzCloud and KEGG databases to detect unknown compounds (mass tolerance: 5 ppm), followed by further screening according to the Human Metabolites Database (HMDB: http://www.hmdb.ca). Most lipids were identified through Lipidsearch 5.1 with default parameters (precursor tolerance: 5 ppm, product tolerance: 8 ppm). In addition, manual additions further enrich metabolite databases in combination with Chemspider results and relevant articles that provided single-cell metabolite identification lists with MS/MS details[21,22,24,56]. The single-cell data were analyzed using the OmicShare online platform for data analysis (https://www.omicshare.com/tools), including data

visualization and clustering by UMAP, t-SNE, KEGG annotation and heatmap. The intensities of diagnostic ions are used for quantifying isomers. The S/N ratio of the diagnostic ions should be >5 for quantitation. The relative ratio of isomers is represented by the ratio of the intensity of a specific diagnostic ion, and the relative amounts of a specific isomer are represented by the intensity of the diagnostic ion divided by the total intensities of all diagnostic ions.

### Fluorescence-based flow cytometry analysis

Fluorescence and bright-field images were acquired using a Nikon Eclipse Ti inverted fluorescence microscope system (Nikon, Japan). FACS were performed using an Astrios EQ flow cytometer (BeckMan Coulter, USA). The sample pretreatment method used was based on the indirect flow cytometry protocol provided by the official abcam website. Briefly, harvested cells (-$10^7$ cells/tube) were fixed with 4% paraformaldehyde solution, followed by permeabilization with cold methanol. Primary and fluorochrome-labeled secondary antibody working solutions, which were both diluted to 1 µg/ml in 3% BSA in PBS, were added consecutively for target protein labeling. After the cells were washed with PBS, they were incubated at 4 °C and analyzed as soon as possible.

### 10xGenomics single cell transcriptome sequencing analysis

A preexperiment utilizing Trypan blue staining was conducted to ensure that the viability of the wtMCF-7 cells was greater than 90% and that the cells were in optimal growth conditions, with a confluency requirement of greater than 80%. Subsequently, the culture medium was washed off using Dulbecco's PBS (DPBS) buffer solution without calcium or magnesium, followed by digestion with 0.25% EDTA-treated trypsin at 37 °C for 3 min. The detached cells were then collected by centrifugation at 500 g for 5 min after being transferred to a centrifuge tube using culture medium. After the cells were resuspended in DPBS buffer solution and cell counting was performed, the cells were diluted to a density of 10e6 per milliliter. Once the cell preparation was completed, the cells were immediately placed on ice for subsequent library construction using the 10x Genomics platform. The 10xGenomics single-cell transcriptome sequencing project was carried out by the BGI sequencing platform. The distribution of the fragment sizes was checked by an Agilent 2100 bioanalyzer, and the final products were sequenced using the DNBSEQTM platform.

The Cell Ranger Software Suite (10x Genomics Cell Ranger 6.1.2) was used for scRNA-seq data processing. The hg38 reference dataset (refdata-gex-GRCh38-2020-A) required for Cell Ranger was obtained from 10x Genomics. Briefly, the sequenced FASTQ files of the MCF-7 sample were aligned to the hg38 human reference genome using STAR software with the Cell Ranger "count" module. Next, a feature-barcode matrix was generated from Cell Ranger. Downstream analysis was performed using the R package Seurat (v 3.2.0). Quality control was applied to cells based on the number of detected genes and the proportion of mitochondrial reads per cell. The average number of detected genes and transcripts (UMI counts) per cell were 1897 and 7072, respectively. (Supplementary Table. 1 and Supplementary Fig 17). Specifically, cells with fewer than 200 detected genes or cells with >90% of the proportion of the maximum genes were filtered out. For the mitochondrial metric, the cells were sorted in descending order of the mitochondrial read ratio, and the top 15% of the cells were filtered out. Potential doublets were identified and removed by the Doublet Detection function. In detail, principal component ($n = 15$) analysis was conducted using only the 2000 highly variable genes in the dataset. UMAP was then used for two-dimensional visualization of the resulting clusters. Differentially expressed genes across different cell clusters were identified using the FindMarkers function in Seurat with the parameters 'logfc. Threshold > 0.25, minPct > 0.1 and Padj ≤ 0.05'. KEGG (V93.0) pathway enrichment analysis was performed with differential expressed genes by hypergeometric test using R function 'phyper', the resulted $P$-values was corrected to Q values by Benjamini−Hochberg method. KEGG pathways with $Q$-value <= 0.05 were considered to be significantly enriched.

### Clustering algorithm

K-means clustering based on Euclidean distance were used for clustering of single cell metabolome data. The best number of clusters was determined by R package NbClust with parameters 'distance = " euclidean", min.nc = 2, max.nc = 15, method = "kmeans", index = "all-long", alphaBeale = 0.1', which provided 30 indices to evaluate the performance of clustering results and varied the number of clusters from 2 to 15 and evaluating these indices at each cluster size to selected optimal number of clusters[57].

In the UMAP analysis of the OmicShare online platform, the UMAP parameters were the same as the default parameters in the UMAP package in R. The main parameters were set as follows: n_neighbors = 15, n_components = 2, metric = "Euclidean", n_epochs = 200, min_dist = 0.1, and negative_sample_rate = 5. UMAP analysis of the BGI platform referred to relevant literature[58]. The main parameters were set as follows: n_neighbors = 15, n_components = 2, metric = " Euclidean", min_dist = 0.2. Principal component analysis (PCA) was employed to conduct tSNE dimensionality reduction on the data, and the main parameters were set as follows: min_dist = 0.4, k = 2, initial_dims = 30, perplexity = 30, max_iter = 1000, min_cost = 0, and epoch = 100.

### Reporting summary

Further information on research design is available in the Nature Portfolio Reporting Summary linked to this article.

## Data availability

Single cell sequencing data that support the findings of this study have been deposited in Gene Expression Omnibus (GEO) with the accession codes GSE262591. The metabolomic MS raw data have been deposited to MetaboLights with the dataset identifier MTBLS8113. The remaining data are available within the article, Supplementary Information or Source Data file. Source data are provided with this paper.

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

## Acknowledgements

This work was financially supported by the National Key R&D Program of China (2022YFC3400700 to Y.B.) and the National Science Fund of China (No. 22125401 and 22074003 to Y.B.). The authors would like to thank Mr. Hongmin Yu from the Fuchou Tang group of Peking University for the help in the analysis of the sequencing data.

## Author contributions

Y.B. proposed the study concept and strategy. S.J.Q., Y.Z. and Y.B. designed and performed the experiments and data analysis. M.Y.S. and D.Y.M. contributed to the instrument construction and experiments. J.S.L. and L.W. contributed to the sequencing data analysis. S.J.Q. and Y.B. wrote the paper.

## Competing interests

The authors declare no competing interests.
