## [Peer Review File · Nature Communications]

Reviewers' Comments:

Reviewer #1:

Remarks to the Author:

The authors have present an interesting approach for single cell evaluation. This approach provides exciting possibilities for metabolic measurements of single cells, an area where few solutions are available. There are a number of clarifications or edits that would improve the manuscript and the clarity of these new methods.

MAJOR

The order of appearance of the Supplementary Figures doesn't seem to match the text, and I think for a few there may not have been a reference in the text itself? Adding or clarifying these would help to ensure the data is linked to the presentation of results.

Figure 2 a: the step signals start and drop sharply at the beginning and end of each lysed cell. I would expect some dilution of the lysate to occur, meaning the front and back end of the lysate would become less concentrated, resulting in a more gentle increase/decrease. Is there an apparent reason it looks this way?

Lineage 114: "After sonication, obvious cell shrunk accompanied with nearly entire disappearance of the staining signal of cell nucleus and membrane was observed, which confirmed the destruction of cell structure (Figure S3)."

-- This statement is critical to method, but is there additional data demonstrating how effective the sonication is? Intact cells that remain may contaminate the signals processed that should indicate lysed cells and convolute the analysis. If lysis was 100% effective then this would be less of an issue, but as indicated, some cells remain.

Line 116-128: concerning the pulse (cells) and step (lysed cell) signals.

-- It seems that intact pulse cells may be occurring within the step signal if lysis is incomplete. Is there a method here to evaluate this?

Line 244-247: "Notably, no obvious clustering within MCF-7 cells were observed with the above metabolite data from single cells (Figure 5b). Conversely, upon the combination of BHB/GHB isomers information, 3 clusters of cells were distinguished clearly."

-- This is a somewhat subjective assessment based on the visual appearance on the UMAP, but should be backed up by a more detailed cluster evaluation. Alternatively, this could simply be compared -- greater cell diversity apparent after the addition of the isomer information for clustering, when compared to clustering without. It seems an overly obsessive point, but while UMAP plots are excellent visualisation tools, it is challenging to make quantitative assessments with them.

Line 269: "isomer ratio (Figure S15), which suggested that the GHB may not contribute to the cell clustering"

-- This is speculative, which is OK, but should be clarified as such.

Line 273: " No significant difference of above enzymes was observed among MCF-7 cells (Figure S16)."

-- Only showing fold changes without statistical evaluation — perhaps use a volcano plot or some other statistical assessment:

-- Similar with Figure 6a and Figure 7 -- fold alone helpful, but some form of statistical comparison important.

Line 433-434: "and the cells was mono-dispersed by ordering capillary twined into 6 loops with diameters of 6 cm."

-- Any data on separation?

Line 506: regarding 10X genomics, more detail of the cell preparation protocols is required.

MINOR

Line 62: "online" -- given the varied readership of the journal, it may be worth clarifying here what online vs offline methods are explicitly.

Line 91 "Single cell metabolite ambiguous identification" -- perhaps rephrase to "identification of ambiguous metabolites in single cells"

Line 147: "which obviously higher than those in other mass cytometry and our previous organic mass cytometry (1592 and 303)"
-- Reference required

Line 171: concerning the ice bath
-- Given the long run time of the instrument, and ice bath would presumably melt and eventually lose temperature regulation. Do you have any data on this, or alternatives for long runs?

Line 426: rpm values should be given in rcf (xg). Again line 504.

Line 441: regarding 10 min cleaning, how was this length determined to be suitable?

Reviewer #2:

Remarks to the Author:

In this study, authors developed a single-cell metabolome platform: in-depth organic mass cytometry (ID-organic cytoMS). This setup used Dean flow for single cell isolation, ice-cold sonication for cell lysis, and non-contact ESI for molecular ionization. This design provided the extended ion signal duration time for MS/MS metabolites' identification and isomers' discrimination with relatively high throughput. In addition, authors conducted single cell transcriptomics analysis. Using combined metabolomics and transcriptomics data, comprehensive studies of cell heterogeneity could be achieved.

The method reported in this manuscript possesses a certain level of novelty. The overall design of the cell sorting, cell lysis, and ionization is relatively simple, and it can be potentially adopted by other research groups for different types of studies. In addition, a relatively high throughput can be achieved. However, there are a few flaws in their studies.

(1) The procedures of cell preparation have flaws. Culture cells were stained, washed by methanol/water (80/20), and then resuspended in the methanol/water solution. All processes added severe stresses on cells and changed cell metabolisms. In addition, this methanol/water solution can extract molecules from cells during these processes, even cells may look intact. These changes may vary for different metabolites and cell lines.

(2) Dean flow-based capillary has been previously used for single cell sorting and mass spectrometry studies (Ref. 30). This current study utilized a similar setup, and the only novelty seems to be the ice bath sonication. The novelty is limited.

(3) The noncontact ESI was used for ionization, but the mechanisms were not fully addressed. It seemed the major purpose of having this design is to reduce clogging issues of the T-junction design. First, it is unnecessary to have the noncontact ESI. In traditional ESI (or nanoESI), the ionization voltage needs to be applied to the sample solution through conductive electrode. There is a metal union connecting the nanoESI emitter and cell sorting capillary (Fig. S1(b)), and the ionization voltage can be directly applied on it. Second, the working mechanisms of this design were unclear. Non-contact ESI can be introduced using AC voltage (inductive electrospray ionization mass spectrometry (iESI-MS)). According to authors, the ionization voltage (from the mass spectrometer) of their noncontact DC ionization device was applied through a copper wire surrounding the fused silica capillary (no coating and no tapering). This seems impractical because the fused silica (with a wall of ~100 um thickness) is not conductive.

(4) The method used to obtain the relative abundances of multiple metabolites has flaws. For example, in Fig. 4 the relative ratios of isomers were reported, but it is unclear how the % numbers were calculated. Did authors simply use the relative abundances of featured MS/MS fragments (highlighted peaks with structures)? This is incorrect without the calibration. The relative abundances of fragments cannot represent the relative abundances of their parent ions, which have different structures. Authors need to run calibration experiments using the prepared solutions of these standard isomers with multiple known concentration ratios (e.g., 0.5:1, 1:1, 1:2, 1:5), and MS/MS need to be performed using the same experimental setup and conditions, particularly the same CID or HCD energy. The relative intensities of the same fragments can be used to establish the calibration line, which can be used to obtain the relative abundances of these isomers in cells.

There are also some minor issues.

(1) The color contrast needs to be increased for plots in Fig. 7a and 7c. Some of those bars in these plots are indistinguishable.

(2) There are some language issues.

Lin 115. The description of "disappearance of the staining signal of cell nucleus and membrane" is unclear. What is staining signal? MS signals of stains or the color of stains?

Line 125-128. "It further confirmed that the pulse signals ... the cell with online sonication." This sentence needs to be revised.

Lin 161-163. It says "The characteristic MS peaks of single-cell increased along with the increase of the methanol ratio in cell suspension (Figure S6)." Did it mean peak intensities increased?

Line 154-155. Two things are unclear in the sentence "...observed with mass resolution of 280000 than that of 35000, which demonstrated that metabolites alignment confidence can be enhanced by accurate m/z". (a) What is "that"? What were the conditions compared here? (b) What is "metabolites alignment"?

Line 218. It says "relative expression ratio of the isomer pairs among...". The expression of genes and proteins was commonly used, but the expression of metabolites is never heard.

Line 340-344. The sentence "Beyond that, ... BHB-mediated signal transduction." needs to be rephrased.

Reviewer #3:

Remarks to the Author:

In this manuscript the authors present a novel platform for single-cell metabolome profiling - in-depth organic mass cytometry (ID-organic cytoMS). The major improvement to the previous versions is the extended single-cell analysis time that improves MS/MS acquisition and thus identification of metabolites and discrimination of isomers. As an example, the authors use single MCF-7 cells, and show that they can identify 213 metabolites in a negative and 381 in a positive mode, respectively, which is an improvement to the performance provided by the other single-cell level metabolome profiling methods. The presented method sounds to be a promising improvement to single-cell metabolome profiling.

In the second part of the manuscript, the authors subtype MCF-7 cells by defining relative abundance of example isomers. They show that relative abundance of 4-hydroxybutanoic acid (GHB) and 3-hydroxybutanoic acid (BHB) isomers varies a lot among cells. Next, single-cell RNA sequencing and FACS assay was used to identify two anti-oxidative stress-related proteins. The validation assay shows that expression of the identified proteins positively correlates with the BHB isomer abundance. The authors conclude that this knowledge may broaden the scope of drug candidates for breast cancer therapy.

In the following, I mostly concentrate on analysis and interpretation the single-cell RNA-seq (scRNA-seq) data, which is also my major concern of this paper. I do understand that the major focus of this paper is on the expression of the GHB and BHB metabolism related enzymes. But that does not rule out the fact that the authors should make a better effort on a proper basic analysis of the scRNA-seq data, and appropriately describe the methods used. In the current version of the manuscript, the authors seem to choose the genes they discuss based on publications without showing the full analysis of the data. Many results are represented in figures showing 'expression change fold', but it remains unclear how the data supports the conclusions. Generally, the

manuscript needs revision for English language and grammar, remove unnecessary repetition and pay attention to the proper use of academic language.

L231-233, Figure 4: "as well as their expression among 40 single cells, respectively. (orange or green highlight MS peak indicates characteristic fragmental ions, relative ratio: the intensity ratio of the orange peak to the green one.)". This part of the Figure 4 legend should be revised to clearly explain the content of the figure.

L246 and Figure 5C: It is unclear from the text was information on all detected isomers used in clustering or only that of the BHB/GHB isomers? To obtain unbiased result, I would have used information on all available isomers. Also, the methods section should include more detailed description on the clustering performed.

L273-276, Figure S16: "No significant difference of above enzymes was observed among MCF-7 cells (Figure S16). Above results suggested that the cluster discrimination were mainly attributed to differential expression of BHB but not GHB."

I think the paper does not show as a proper proof for the contribution of BHB and not GHB on the clustering of MCF-7 cells. "The expression change fold" measure used does not tell anything about the fraction of the cells expressed or the expression level itself. It only tells how much the expression of the chosen gene differs from the average expression. It is also not a statistical test, that's why it is misleading to conclude: "No significant difference of above enzymes was observed among MCF-7 cells (Figure S16)".

For the candidate genes, it would have been better to show feature plots for all cells together with Vln or boxplots of normalized expression values, and a barplot showing a % of cells expressed. That would a better illustrate the contribution of the GHB. Generally, unbiased analysis e.g., to identify the most variable genes, or genes that mostly contribute to clustering (if clustering is used in the analysis), could help to interpret the data.

L291-305: This section discusses about the two genes MT2A and hnRNP A1 chosen for the additional analysis. Again, the unbiased analysis of the whole scRNA-seq data would have been needed to show why these two genes were good candidates for the additional analysis. The manuscript states that: "Single cell sequencing results showed the most prominent differential MT2A expression comparing control gene of ACTB", but it is unclear how the test between each gene and the control gene has been done and what was the result. The type of upper Figures 6A shows indeed that there is a group of cells that are higher expressed in MT2A than in the control gene, but the figure fails to show the full expression distribution of MT2A.

L306-307: "Interestingly, we found that expression of MT2A was increased segmentally with P53 expression decline (Figure S17)". The results of the paper do not properly support this conclusion. Again, proper comparison, e.g., with correlation between MT2A and p53 gene expression could be a solution.

L309-312: "In addition, we observed significant expression change-fold of other BHB target proteins relating to anti-oxidative stress as well as aging progress, including SOD[40, 48-49], SULT2B1[50-51], GATA3[52], PTEN[45] etc (Figure S18)". The same comment than above. The Figure S18 does not proof the conclusion made.

L479: the authors use the OmicShare online platform for data analysis. To make the review process easier, in the future I would encourage authors to use more commonly used data analysis platforms with verified scientific publication or direct programming with e.g., R or Python.

L480-482: "data visualized and clustering by Uniform Manifold Approximation and Projection (UMAP)". The authors write that they used UMAP for clustering and visualization. UMAP is a method for dimensional reduction that can indeed be used for visualization of clustering, but it's not a clustering method itself. For clustering, a proper data clustering method should be used and described in the method. The authors use the term "KEGG analysis" instead of enrichment analysis. I think the methods section needs to be written more carefully.

L506-509: "10xGenomics single cell transcriptome sequencing project was undertaken by BGI sequencing platform, which covers the steps of cell filter, library construction, sequencing and data analysis, including cell clustering, gene statistics and enrichment analysis etc."

Instead of only explaining that 10X sequencing was performed by BGI sequencing platform, the authors should follow the instructions of the journal and commonly used procedure to describe in detail, single-cell samples, library preparation, sequencing, and the exact methods, software and thresholds used for pre-processing, quality control and analysis. Now this all remains unclear to the reader and makes it difficult to evaluate the results.

Supplementary Information:

Figure S14: The Figure legend needs revision: 1) Volcano plot has nothing to do with correlation. 2) in this figure metabolite signals were tested between clusters not within clusters. 3) It is unclear why P value limit $P < 0.01$ was chosen, and are the P values corrected for multiple comparison, also it is not defined that $-\log_{10}$ transformation was used for P values. 4) The last sentence "No differential compounds (T test, $P < 0.01$; $|FC| > 1.5$) were depicted in orange and blue dots" sounds weird to me (you are describing what the figure does not show?).

Figure S19: How were the genes chosen for the enrichment analysis? Due to the scale of the Q-value from 0 to 0.5, it is unclear which KEGG pathways are significantly enriched.

Table S3: How were the marker genes chosen?

Manuscript Number: NCOMMS-23-27831

Dear Editors and Reviewers:

We gratefully acknowledge you for the comments concerning our manuscript entitled “In-depth organic mass cytometry reveals differential distribution of 3-hydroxybutanoic acid on single cell level” (No. NCOMMS-23-27831). Those comments are all valuable and very helpful for revising and improving our paper. We tried our best to perform complementary experiments during the past months to improve our work and response reviewers’ comments. The point-to-point responses to the reviewers’ comments are listing below.

Reviewer #1

The authors have presented an interesting approach for single cell evaluation. This approach provides exciting possibilities for metabolic measurements of single cells, an area where few solutions are available. There are a number of clarifications or edits that would improve the manuscript and the clarity of these new methods.

Response:

Thank you very much for your positive evaluation on our work. Following your suggestions, we have not only improved the design of the platform but also reorganized and rewritten the whole manuscript carefully to clearly demonstrate our work.

1. *The order of appearance of the Supplementary Figures doesn't seem to match the text, and I think for a few there may not have been a reference in the text itself? Adding or clarifying these would help to ensure the data is linked to the presentation of results.*

Response: Thank you very much for your suggestions and sorry for our carelessness. We have carefully revised the whole manuscript. The order of Supplementary Figures and Tables have been rearranged. Reference clarifications and corrections have been made in the revised manuscript. (see page 5, line 98; page 6, line 113; page 8, line 164; page 9, line 189)

2. *Figure 2a: The step signals start and drop sharply at the beginning and end of each lysed cell. I would expect some dilution of the lysate to occur, meaning the front and back end of the lysate*

would become less concentrated, resulting in a more gentle increase/decrease. Is there an apparent reason it looks this way?

Response: Thank you very much for your questions. The step signal actually has been bothering us all along in the previous device using MeOH as the cell suspension solution. Following you and other reviewers' comment, in the past several months, we have been re-examining that platform, redesigning and constructing it to solve the potential problem. In this new device, the morphology of cell extension pulses indeed conforms to your proposal, following a normal distribution. It is noteworthy that the final methanol concentration remains basically unchanged when compared with previous configuration, but the cells are introduced by suspending in ammonium formate, ordering in capillary, mixing with methanol sheath liquid, lysing with the help of ultrasonication, then detecting by non-contacted ESI-MS. That means the duration of cells kept in MeOH is much shorter than before, which may contribute to the changing of signal shape. We are also interested in this phenomenon and may try dynamic simulation in the future work.

3. *Lineage 114: "After sonication, obvious cell shrunk accompanied with nearly entire disappearance of the staining signal of cell nucleus and membrane was observed, which confirmed the destruction of cell structure (Figure S3)."*

-- This statement is critical to method, but is there additional data demonstrating how effective the sonication is? Intact cells that remain may contaminate the signals processed that should indicate lysed cells and convolute the analysis. If lysis was 100% effective then this would be less of an issue, but as indicated, some cells remain.

Response: Thank you very much for your valuable suggestions. Apart from observing the morphological changes of cells before and after ultrasonication using fluorescence imaging (as shown in Supplementary Figure 2), we continuously monitored the extended detection performance of 100 cells in mass spectrometry analysis to evaluate cell lysis efficiency, as depicted in figure below or Figure 2 in the main text. By examining the extracted ion chromatogram of characteristic lipid PC (34:1), we determined the extended analysis time for each cell, which ranged from 18 to 30 seconds. This time duration is notably longer compared to existing methods without ultrasonication (~0.6 seconds), indicating that this method exhibits excellent and universal cell lysis efficiency. It takes about 5 min. for each cell to pass through

the ultrasonication region, which supposes to be enough for the lysis of cell. If cells are not 100% lysed, short signal extension or pulsed signal should be obtained, which hasn't been found in experiments. Large amount of cell typing results show the reproducibility of the platform, which proved laterally that cells have been efficiently lysed. Our method has identified the highest number of metabolites among online mass cytometry techniques to date.

4. *Line 116-128: concerning the pulse (cells) and step (lysed cell) signals.*

-- It seems that intact pulse cells may be occurring within the step signal if lysis is incomplete.

Is there a method here to evaluate this?

Response: Thank you very much for your comments. As mentioned in the previous question, we employed fluorescence imaging and mass spectrometry analysis to evaluate the effectiveness of cell lysis with our method. Both approaches indicated that our method can achieve relatively complete cell lysis and maintain excellent performance during long-term continuous injection. On the other hand, there is extremely low chance that intact cell appears in the step signal since cells have been mono-dispersed during sampling step and the capillary is intact with low chance of cell adsorption and clogging. If very low amount of intact cells remain and come out together with the extension signal, one can easily distinguish them by the abnormal TIC intensity and discard them.

5. *Line 244-247: "Notably, no obvious clustering within MCF-7 cells were observed with the above metabolite data from single cells (Figure 5b). Conversely, upon the combination of BHB/GHB isomers information, 3 clusters of cells were distinguished clearly."*

-- This is a somewhat subjective assessment based on the visual appearance on the UMAP, but should be backed up by an more detailed cluster evaluation. Alternatively, this could simply be compared -- greater cell diversity apparent after the addition of the isomer information for clustering, when compared to clustering without. It seems an overly obsessive point, but while UMAP plots are excellent visualisation tools, it is challenging to make quantitative assessments with them.

Response: Thank you for your valuable comments. Firstly, our work demonstrates that cellular diversity significantly increases following the incorporation of isomer information. Furthermore, a reanalysis of the single-cell metabolomics data using the Nbclust package in R have been

conducted, which allowed us to determine the optimal number of clusters under unsupervised conditions (see supplementary figure 13a). The Nbclust package provides multiple evaluation criteria and supports the highest number of indices when there are three clusters. These results suggested that three is indeed the optimal number of clusters. Furthermore, by calculating within-cluster sum of squares (wss) (see supplementary figure 13b), we observed a rapid decrease in wss values from one to three clusters followed by a significant slowdown beyond three clusters, which further confirms our conclusion that cells have been classified into three subtypes.

6. *Line 269: "isomer ratio (Figure S15), which suggested that the GHB may not contribute to the cell clustering"*

-- This is speculative, which is OK, but should be clarified as such.

Response: Thank you for your valuable feedback. We integrated results from different experiments to speculate that GHB may not contribute to the cell clustering. In the transcription levels, our single-cell sequencing data didn't present any significant differences of key enzymes associated with GHB synthesis, transport, and conversion among the three cell subtypes. Western blot assay also confirmed that no significant difference in protein expression level has been found (see supplementary figure 18). On the other hand, notable variations in the expression levels of downstream target proteins regulated by BHB have been confirmed. Therefore, we hypothesize that the disparity in isomer ratio primarily arises from differential BHB expression. All these results have been incorporated into revised main text along with additional explanations to enhance clarity and precision (see page 12 line 268-285).

7. *Line 273: " No significant difference of above enzymes was observed among MCF-7 cells (Figure S16)."-- Only showing fold changes without statistical evaluation — perhaps use a volcano plot or some other statistical assessment:*

-- Similar with Figure 6a and Figure 7 -- fold alone helpful, but some form of statistical comparison important.

Response: Thank you for your valuable suggestions. Following your suggestion, we employed box plots to statistically illustrated our analysis results. Furthermore, we also conducted WB experiment to validate and support our conclusion. (see supplementary figure 17 and 18)

8. *Line 433-434: "and the cells was mono-dispersed by ordering capillary twined into 6 loops with*

diameters of 6 cm."

-- Any data on separation?

Response: Thank you for your valuable question. The dispersion of single cells has been demonstrated in the supplementary information (see supplementary figure 2). In ammonium formate solution, cells predominantly exist as individual entities and maintain their dispersed state upon entering the separation capillary. The mass spectrometry results also yield independent pulse signals, with a pulse count similar to the cell count within the capillary (see supplementary figure 4), thereby confirming excellent monodispersity of the cells and establishing that each pulse corresponds to a single-cell event. Details relating to cell injection and ordering system refer to the method section in the main text. (see page 19 line 457-470)

9. *Line 506: regarding 10X genomics, more detail of the cell preparation protocols is required.*

Response: Thank you for your valuable question. The detailed protocol for preparing cell samples for 10X genomics sequencing experiments has been included in the main text (see page 22 line 535-543). Briefly, wild-type MCF7 cells (with confirmed cell viability >90%) were enzymatically dissociated into single-cell suspension, followed by centrifugation to remove the supernatant. Subsequently, the cells were resuspended in a calcium- and magnesium-free PBS solution, and library construction was promptly initiated thereafter.

10. *Line 62: "online" -- given the varied readership of the journal, it may be worth clarifying here what online vs offline methods are explicitly.*

Response: Thank you for your valuable suggestion. Following your suggestion, considering the varied readership of the journal, we have added the necessary description on off-line and online single cell metabolomics into the main text. (see page 3 line 41-44, page 4, line 59-61)

11. *Line 91 "Single cell metabolite ambiguous identification" -- perhaps rephrase to "identification of ambiguous metabolites in single cells"*

Response: Thank you for your valuable suggestion. We have corrected it (see page 5 line 93). The entire article has undergone a comprehensive grammar check and the writing has been improved.

12. *Line 147: "which obviously higher than those in other mass cytometry and our previous organic mass cytometry (1592 and 303)"-- Reference required*

Response: Thank you for your suggestion and sorry for our carelessness. Due to changes in

the content and the logical progression of the context, we have removed this section in the revised manuscript.

- 13.** *Line 171: concerning the ice bath. Given the long run time of the instrument, and ice bath would presumably melt and eventually lose temperature regulation. Do you have any data on this, or alternatives for long runs?*

Response: Thank you for your valuable suggestion. We agree with you that prolonged ultrasonication generates a substantial amount of heat, resulting in an elevation of the water bath temperature. In order to consistently maintain the ice-water bath at 0 degrees Celsius, we periodically drain excess water and replenish it with fresh ice cubes every 15 minutes. Furthermore, continuous monitoring of the water bath temperature using a thermometer ensures that ultrasound is conducted at a consistently low temperature throughout extended mass spectrometry data acquisition.

- 14.** *Line 426: rpm values should be given in rcf (xg). Again line 504.*

Response: Thank you for your valuable suggestion. We have corrected them.

- 15.** *Line 441: regarding 10 min cleaning, how was this length determined to be suitable?*

Response: Thank you for your valuable suggestion. 10 minutes cleaning time between runs came from our extensive experimental experience. Data collected after 10 min cleaning presented an absence of cell pulses in TIC and the mass spectrum closely resembled that of the blank background, indicating thorough pipeline cleaning at this stage.

Reviewer #2

In this study, authors developed a single-cell metabolome platform: in-depth organic mass cytometry (ID-organic cytoMS). This setup used Dean flow for single cell isolation, ice-cold sonication for cell lysis, and non-contact ESI for molecular ionization. This design provided the extended ion signal duration time for MS/MS metabolites' identification and isomers' discrimination with relatively high throughput. In addition, authors conducted single cell transcriptomics analysis. Using combined metabolomics and transcriptomics data, comprehensive studies of cell heterogeneity could be achieved.

The method reported in this manuscript possesses a certain level of novelty. The overall design of the cell sorting, cell lysis, and ionization is relatively simple, and it can be potentially adopted by other research groups for different types of studies. In addition, a relatively high throughput can be

achieved. However, there are a few flaws in their studies.

Response: Thank you very much for your valuable comments. After carefully and seriously considering your questions and suggestions, we have reconstructed the “In-depth organic mass cytometry” to solve those flaws you mentioned and improve its performance to make it more practical and valuable platform during the past several months.

1. *The procedures of cell preparation have flaws. Culture cells were stained, washed by methanol/water (80/20), and then resuspended in the methanol/water solution. All processes added severe stresses on cells and changed cell metabolisms. In addition, this methanol/water solution can extract molecules from cells during these processes, even cells may look intact. These changes may vary for different metabolites and cell lines.*

Response: Thank you for your valuable and significant feedback. After carefully and seriously considering your questions, to preserve the intrinsic state and structural integrity of the cells, we replaced the methanol/water suspension solution with an isotonic ammonium formate solution according to literatures^[1-2]. Unfortunately, ammonium formate only is not enough to recover the metabolite signal and no sufficient MS signal can be obtained. To increase the extraction of the cellular content, ensure the stable ionization spray during the analysis, minimize potential impact on MS analysis, methanol was introduced as the sheath fluid, followed by subsequent ultrasonication. Ice-bath sonication in non-contact manner is utilized to prevent from the interference of the external energy and degradation of endogenous compounds. We believe this method offers a robust tool for conducting comprehensive and in-depth research on single-cell metabolomics.

2. *Dean flow-based capillary has been previously used for single cell sorting and mass spectrometry studies (Ref. 30). This current study utilized a similar setup, and the only novelty seems to be the ice bath sonication. The novelty is limited.*

Response: Thank you for your valuable suggestions. Single-cell metabolomics holds significant importance; however, the current methods remain highly limited, offering restricted metabolic information. That’s why Nature Publishing Group has recognized single-cell metabolomics technology as a key technology to watch in 2023. We believe it can be partially addressed through integration of various essential technologies. Therefore, we have further enhanced our method and employed a straightforward and accessible approach - utilizing Dean

flow microfluidic separation to realize individual cell ordering while preserving their original state for high-throughput single-cell metabolomics analysis. In comparison to previous online methods, our approach extends analysis time by 40 times increase and yields abundant MS/MS spectra for identifying nearly 600 metabolites, representing the highest number of identifications based on MS/MS currently attainable.

- 3. The noncontact ESI was used for ionization, but the mechanisms were not fully addressed. It seemed the major purpose of having this design is to reduce clogging issues of the T-junction design. First, it is unnecessary to have the noncontact ESI. In traditional ESI (or nanoESI), the ionization voltage needs to be applied to the sample solution through conductive electrode. There is a metal union connecting the nanoESI emitter and cell sorting capillary (Fig. S1(b)), and the ionization voltage can be directly applied on it. Second, the working mechanisms of this design were unclear. Non-contact ESI can be introduced using AC voltage (inductive electrospray ionization mass spectrometry (iESI-MS)). According to authors, the ionization voltage (from the mass spectrometer) of their noncontact DC ionization device was applied through a copper wire surrounding the fused silica capillary (no coating and no tapering). This seems impractical because the fused silica (with a wall of ~100 um thickness) is not conductive.*

Response: Thank you for your valuable suggestions. The enhanced experimental setup has been proposed to exclude a metal union connecting the nanoESI emitter and cell sorting capillary. In response to your remaining inquiries, achieving electrospray by applying high voltage on the metal tee is indeed feasible; however, this approach necessitates additional high-voltage generation equipment and does not allow for convenient switching between positive and negative modes in the mass spectrometry data acquisition software, potentially impacting its flexibility and simplicity. In contrast, non-contact electrospray only requires a nano-electrospray ion source, rendering it relatively simpler with enhanced safety and more convenient mode switching between positive and negative polarities. Furthermore, by minimizing the use of two-way connectors, it reduces the occurrence of cell clogging situations. Secondly, non-contact electrospray shares fundamental similarities with induced electrospray in principle^[3-4] but diverges in utilizing copper wire wrapped around the outer wall of a burned-off coated capillary instead of a metal plate to generate an induced electric field. Additionally, while induced electrospray employs a high-voltage alternating current power supply, non-

contact electrospray solely necessitates provision of high-voltage direct current. The capillary initially acts as an insulator prior to coating removal but acquires certain conductivity thereafter; thus inducing a high-voltage electric field at its tip through external copper wire to facilitate ionization. In our experiment, an uncoated capillary with dimensions of 360 μm outer diameter and 150 μm inner diameter could be served as a spray needle for ionization.

4. *The method used to obtain the relative abundances of multiple metabolites has flaws. For example, in Fig. 4 the relative ratios of isomers were reported, but it is unclear how the % numbers were calculated. Did authors simply use the relative abundances of featured MS/MS fragments (highlighted peaks with structures)? This is incorrect without the calibration. The relative abundances of fragments cannot represent the relative abundances of their parent ions, which have different structures. Authors need to run calibration experiments using the prepared solutions of these standard isomers with multiple known concentration ratios (e.g., 0.5:1, 1:1, 1:2, 1:5), and MS/MS need to be performed using the same experimental setup and conditions, particularly the same CID or HCD energy. The relative intensities of the same fragments can be used to establish the calibration line, which can be used to obtain the relative abundances of these isomers in cells.*

Response: Thank you for your valuable suggestions. According to the major literatures of single cell metabolomics^[2, 5], in order to achieve relative quantification of single-cell metabolites, we first ensure the stability of the instrument and apparatus to guarantee a consistent baseline. Next, we normalize the total ion current (TIC) of cell event-corresponding mass spectrum to eliminate the influence of baseline fluctuations on metabolite quantification. The principle for relative quantification of isomers is similar to classical Selected Reaction Monitoring (SRM), where the relative abundance of characteristic fragment ions is utilized for quantifying parent ions. Since fragment ions have identical fragmentation environments in the same tandem mass spectrum, it becomes possible to compare the intensities of characteristic fragment ions among a group of structural isomers and perform relative quantification based on this information. We have also consulted relevant literature regarding this approach^[6], in which relative quantitation of the lipid C=C location isomers (such as PC 16:0_18:1) was realized using the corresponding relative intensities of specific diagnostic ions. Absolute quantification of isomers can be further achieved through an external standard method using solutions with

different concentrations of standard substances when the amount of sample is sufficient. However, due to limitations such as cellular heterogeneity, complex matrix interference and tiny cellular contents, it remains challenging especially in single cell analysis.

5. *The color contrast needs to be increased for plots in Fig. 7a and 7c. Some of those bars in these plots are indistinguishable.*

Response: Thank you for your valuable suggestions. We have improved it. (see page 16 Figure 7)

6. *Lin 115. The description of “disappearance of the staining signal of cell nucleus and membrane” is unclear. What is staining signal? MS signals of stains or the color of stains?*

Response: Thank you for your comments and sorry for the unclear expression. We want to express that, the loss of green fluorescence (excitation light from cell membrane dye) and blue fluorescence (excitation light from cell nucleus dye) was observed under a fluorescent microscope. We have made the corresponding revisions. (see page 6 line 118-121)

7. *Line 125-128. “It further confirmed that the pulse signals ... the cell with online sonication.” This sentence needs to be revised.*

Response: Thank you for your comments. Since we have reconstructed the platform, relevant content has been removed from the manuscript.

8. *Lin 161-163. It says “The characteristic MS peaks of single-cell increased along with the increase of the methanol ratio in cell suspension (Figure S6).” Did it mean peak intensities increased?*

Response: Thank you for your valuable questions. The increase in methanol proportion facilitates the extraction of metabolites, leading to a gradual substitution of dominant peaks in the spectrum with cellular metabolites and the enhanced intensity of endogenous metabolites. Since we have reconstructed the platform, methanol proportion was not further be optimized, therefore relevant content has been removed.

9. *Line 154-155. Two things are unclear in the sentence “...observed with mass resolution of 280000 than that of 35000, which demonstrated that metabolites alignment confidence can be enhanced by accurate m/z”. (a) What is “that”? What were the conditions compared here? (b) What is “metabolites alignment”?*

Response: Thank you for your valuable questions and sorry for our unclear sentences. We have

corrected it as following, “Taking the MS peak of m/z 130.0868 (leucine/Isoleucine) as an example, mass resolution of 280000 offers definitely narrow half-width of MS peak than mass resolution of 35000, which enhance the confidence in metabolite assignment.” (see page 8 line 156-159).

10. *Line 218. It says “relative expression ratio of the isomer pairs among...”. The expression of genes and proteins was commonly used, but the expression of metabolites is never heard.*

Response: Thank you for your valuable comments. In relation to the depiction of metabolite content, we have conducted a comprehensive examination and made revisions throughout the entire article, substituting “expression ratio” with “abundance ratio”. (see page 10 line 219, page 12 line 263, page 16 line 353)

11. *Line 340-344. The sentence “Beyond that, ... BHB-mediated signal transduction.” needs to be rephrased.*

Response: Thank you for your valuable suggestions. The statement “Beyond that, ... BHB-mediated signal transduction.” has been reformulated as “Furthermore, several observed key marker genes involved in transcription regulation, stress response, and neurodevelopmental disorders (Supplementary Fig. 23 and Supplementary Table 1), including KCNQ1OT154, NEAT1 and histone encoding genes HIST1H1B and HIST1H4C, may also play dispensable roles in BHB-mediated signal transduction.”. (see page 15 line 343-347)

Reviewer #3

I do understand that the major focus of this paper is on the expression of the GHB and BHB metabolism related enzymes. But that does not rule out the fact that the authors should make a better effort on a proper basic analysis of the scRNA-seq data, and appropriately describe the methods used. In the current version of the manuscript, the authors seem to choose the genes they discuss based on publications without showing the full analysis of the data. Many results are represented in figures showing ‘expression change fold’, but it remains unclear how the data supports the conclusions. Generally, the manuscript needs revision for English language and grammar, remove unnecessary repetition and pay attention to the proper use of academic language.

Response: Thank you very much for your comments. Through effective communication and deep learning in the field of single-cell sequencing, we have successfully conducted a comprehensive

analysis and presented the existing single-cell sequencing data in a meticulous manner. Furthermore, we have enhanced the experimental methods for single-cell sample preparation, library construction, and sequencing. In order to convey our findings more accurately, we have replaced the concept of “expression change fold” with appropriate statistical terminology. Additionally, we have diligently reviewed the entire paper to rectify any English grammar issues and ensure precise usage of academic language. Specifically addressing your queries, we made the following modifications:

1. *Line 231-233, Figure 4: “as well as their expression among 40 single cells, respectively. (orange or green highlight MS peak indicates characteristic fragmental ions, relative ratio: the intensity ratio of the orange peak to the green one.)”.* This part of the Figure 4 legend should be revised to clearly explain the content of the figure.

Response: Thank you very much for your suggestions. We have made further revisions to the caption of Figure 4 for better interpretation as follows: “... Relative abundance distribution among 60 single cells was shown below, respectively. (characteristic fragmental ions and corresponding MS peak were highlighted with orange or green color, isomers relative ratio was defined as the intensity ratio between the characteristic fragmental ions.) ” (see page 11 line 230-233)

2. *L246 and Figure 5C: It is unclear from the text was information on all detected isomers used in clustering or only that of the BHB/GHB isomers? To obtain unbiased result, I would have used information on all available isomers. Also, the methods section should include more detailed description on the clustering performed.*

Response: Thank you very much for your suggestions. We assessed the capacity to discriminate between cell subtypes obtained from three pairs of isomers and observed that only the GHB/BHB pair significantly enhanced differentiation of MCF7 cells. (see supplementary figure 14). Consequently, we exclusively incorporated information on this particular pair of isomers in subsequent analyses, which also implies their distinctive role in MCF7 cells. The clustering algorithm employed and parameter settings have been provided as method section. (see page 21 line 513-517)

3. *L273-276, Figure S16: “No significant difference of above enzymes was observed among MCF-7 cells (Figure S16). Above results suggested that the cluster discrimination were mainly attributed to differential expression of BHB but not GHB.” I think the paper does not show as*

a proper proof for the contribution of BHB and not GHB on the clustering of MCF-7 cells. “The expression change fold” measure used does not tell anything about the fraction of the cells expressed or the expression level itself. It only tells how much the expression of the chosen gene differs from the average expression. It is also not a statistical test, that’s why it is misleading to conclude: “No significant difference of above enzymes was observed among MCF-7 cells (Figure S16)”.

Response: Thank you very much for your suggestions. We have reorganized the sequencing data and established a correlation between the three identified cell clusters in the sequencing results and single-cell metabolomics outcomes. By employing box plots to compare the expression levels of GHB-related enzymes across different cell clusters (supplementary figure 17), we confirmed no significant differences in their expression levels among these three cell clusters. Furthermore, our findings were validated at the protein level through Western blot experiments on FACS-sorted cells, yielding consistent results. (see supplementary figure 18)

4. *For the candidate genes, it would have been better to show feature plots for all cells together with Vln or boxplots of normalized expression values, and a barplot showing a % of cells expressed. That would a better illustrate the contribution of the GHB. Generally, unbiased analysis e.g., to identify the most variable genes, or genes that mostly contribute to clustering (if clustering is used in the analysis), could help to interpret the data.*

Response: Thank you very much for your suggestions. For the candidate genes, we have showed feature plots for all cells together with boxplots of normalized expression values, and a barplot showing a % of cells expressed. (see figure 6b and supplementary figure 19, 20). Analysis of differential gene expression among distinct cell clusters (see supplementary Table 1), along with comparison to MCF7 single cell sequencing-related literature has been supplemented in the main text. (see supplementary figure 16)

5. *L291-305: This section discusses about the two genes MT2A and hnRNP A1 chosen for the additional analysis. Again, the unbiased analysis of the whole scRNA-seq data would have been needed to show why these two genes were good candidates for the additional analysis. The manuscript states that: “Single cell sequencing results showed the most prominent differential MT2A expression comparing control gene of ACTB”, but it is unclear how the test between each gene and the control gene has been done and what was the result. The type of upper Figures 6A*

shows indeed that there is a group of cells that are higher expressed in MT2A than in the control gene, but the figure fails to show the full expression distribution of MT2A.

Response: Thank you very much for your suggestions. According to the research findings from single-cell metabolomics, we have observed a differential abundance distribution of BHB in MCF-7 cells, indicating substantial cellular heterogeneity within BHB-related pathways. Consequently, our study focused on investigating the protein expression associated with BHB in individual MCF-7 cells to validate our conclusions. Based on existing literature^[7-8], we identified MT2A and hnRNP A1 as downstream target proteins closely linked to BHB levels. Moreover, global sequencing analysis also revealed significant variations in the expression of these two proteins across different cell clusters. To statistically analyze specific results, box plots were employed, which can be seen in figure 6 in the main text (page 14)

6. *L306-307: “Interestingly, we found that expression of MT2A was increased segmentally with P53 expression decline (Figure S17)”. The results of the paper do not properly support this conclusion. Again, proper comparison, e.g., with correlation between MT2A and p53 gene expression could be a solution.*

Response: Thank you very much for your suggestions. After conducting supplementary experiments and thorough data analysis during last several months, we have discovered a segmental increase in the expression of MT2A as P53 expression declines. This phenomenon also helps us further understand the potential regulatory role between MT2A and P53. As P53 is not the key this study, related discussion has been removed. However, this phenomenon is worth exploring in depth in future research endeavors.

7. *L309-312: “In addition, we observed significant expression change-fold of other BHB target proteins relating to anti-oxidative stress as well as aging progress, including SOD[40, 48-49], SULT2B1[50-51], GATA3[52], PTEN[45] etc (Figure S18)”. The same comment than above. The Figure S18 does not proof the conclusion made.*

Response: Thank you very much for your suggestions. Similar to the previous inquiry, we performed statistical analysis using box plots to assess the expression of BHB-related target proteins, revealing significant variations in gene expression across distinct cell clusters. (see supplementary figure 21)

8. *L479: the authors use the OmicShare online platform for data analysis. To make the review*

process easier, in the future I would encourage authors to use more commonly used data analysis platforms with verified scientific publication or direct programming with e.g., R or Python.

Response: Thank you very much for your useful suggestions. We have to say that omicsshare is an online bioinformatics analysis platform with the R language, which is user friendly to people who are not specialized in in bioinformatics analysis. We will sincerely consider your suggestion in our future work.

9. *L480-482: “data visualized and clustering by Uniform Manifold Approximation and Projection (UMAP)”. The authors write that they used UMAP for clustering and visualization. UMAP is a method for dimensional reduction that can indeed be used for visualization of clustering, but it’s not a clustering method itself. For clustering, a proper data clustering method should be used and described in the method. The authors use the term “KEGG analysis” instead of enrichment analysis. I think the methods section needs to be written more carefully.*

Response: Thank you very much for your valuable suggestions. Integrating the comments from several reviewers, we have conducted a reanalysis of the single-cell metabolomics data using the Nbclust package and calculation of within-cluster sum of squares (wss) in R, which allowed us to determine the optimal number of clusters under unsupervised conditions (see supplementary figure 13). Relating description has been supplemented in the method section. KEGG analysis was utilized for metabolites pathway annotation and differential analysis, which has replaced relating term in main text. (see page 9 line 196, page 21 line 513-518)

10. *L506-509: “10xGenomics single cell transcriptome sequencing project was undertaken by BGI sequencing platform, which covers the steps of cell filter, library construction, sequencing and data analysis, including cell clustering, gene statistics and enrichment analysis etc.” Instead of only explaining that 10X sequencing was performed by BGI sequencing platform, the authors should follow the instructions of the journal and commonly used procedure to describe in detail, single-cell samples, library preparation, sequencing, and the exact methods, software and thresholds used for pre-processing, quality control and analysis. Now this all remains unclear to the reader and makes it difficult to evaluate the results.*

Response: Thank you very much for your valuable suggestions. We have provided a comprehensive workflow, encompassing sample preparation and data analysis, in the methods

section for single-cell sequencing experiments (see page 22 line 536-561). Overall, the quality control measures implemented for both single-cell samples and sequencing results demonstrate high reliability.

11. *Figure S14: The Figure legend needs revision: 1) Volcano plot has nothing to do with correlation. 2) in this figure metabolite signals were tested between clusters not within clusters. 3) It is unclear why P value limit $P < 0.01$ was chosen, and are the P values corrected for multiple comparison, also it is not defined that $-\log_{10}$ transformation was used for P values. 4) The last sentence “No differential compounds (T test, $P < 0.01$; $|FC| > 1.5$) were depicted in orange and blue dots” sounds weird to me (you are describing what the figure does not show?).*

Response: Thank you very much for your valuable suggestions and sorry for our carelessness. The figure legend has been revised carefully (see supplementary figure 15). Specially, P value limit was set 0.05, and $-\log_{10}$ transformation was used for P values. The P values are not corrected for multiple comparison. For “No differential compounds (T test, $P < 0.01$; $|FC| > 1.5$) were depicted in orange and blue dots”, we want to express that No significant difference of metabolites abundance occurred among clusters, and the relevant description of supplementary figure 15 has been clarified again.

12. *Figure S19: How were the genes chosen for the enrichment analysis? Due to the scale of the Q-value from 0 to 0.5, it is unclear which KEGG pathways are significantly enriched.*

Response: Thank you very much for your valuable suggestions. KEGG pathways enrichment analysis first maps all candidate genes to each entry in the Gene Ontology database (<http://www.geneontology.org/>), calculates the number of genes per entry, and then applies a hypergeometric test to find the number of genes that are significantly enriched in candidate genes compared to all background genes of the species. We used R's basis function phyper (<https://stat.ethz.ch/R-manual/R-devel/library/stats/html/Hypergeometric.html>) to calculate P value. The P value was then corrected by multiple testing, and the correction package was q value (<https://bioconductor.org/packages/release/bioc/html/qvalue.html>). Finally, Q value (corrected P value) ≤ 0.05 was used as the threshold, and pathways with a final Q value ≤ 0.05 are defined as those significantly enriched in differentially expressed genes, therefore oxidative phosphorylation, thermogenesis and Alzheimer disease related pathways were significantly enriched.

13. *Table S3: How were the marker genes chosen?*

Response: Thank you very much for your question. According to single-cell sequencing result, marker genes were selected among cell clusters with differentially significant expression. Detailed marker genes selection method was supplemented in method section. (see page 23 line 557-561)

References:

- [1] Y. Shao, Y. Zhou, Y. Liu, W. Zhang, G. Zhu, Y. Zhao, Q. Zhang, H. Yao, H. Zhao, G. Guo, S. Zhang, X. Zhang, X. Wang, Intact living-cell electrolaunching ionization mass spectrometry for single-cell metabolomics, *Chem. Sci.* 13 (2022) 8065-8073.
- [2] H. Yao, H. Zhao, X. Zhao, X. Pan, J. Feng, F. Xu, S. Zhang, X. Zhang, Label-free Mass Cytometry for Unveiling Cellular Metabolic Heterogeneity, *Anal. Chem.* 91 (2019) 9777-9783.
- [3] G. Huang, G. Li, R. G. Cooks, Induced Nano-electrospray Ionization for Matrix-Tolerant and High-Throughput Mass Spectrometry, *Angew. Chem. Int. Ed.* 50 (2011) 9907-9910.
- [4] H. Zhu, G. Zou, N. Wang, M. Zhuang, W. Xiong, G. Huang, Single-neuron identification of chemical constituents, physiological changes, and metabolism using mass spectrometry, *Proc Natl Acad Sci U S A* 114 (2017) 2586-2591.
- [5] Q. Liu, W. Ge, T. Wang, J. Lan, S. Martinez-Jarquin, C. Wolfrum, M. Stoffel, R. Zenobi, High-throughput Single-cell Mass Spectrometry Reveals Abnormal Lipid Metabolism in Pancreatic Ductal Adenocarcinoma, *Angew. Chem., Int. Ed. Engl.* (2021)
- [6] Z. Li, S. Cheng, Q. Lin, W. Cao, J. Yang, M. Zhang, A. Shen, W. Zhang, Y. Xia, X. Ma, Z. Ouyang, Single-cell lipidomics with high structural specificity by mass spectrometry, *Nat. Commun.* 12 (2021) 2869-2869.
- [7] T. Shimazu, M. D. Hirschey, J. Newman, W. He, K. Shirakawa, N. L. Moan, C. A. Grueter, H. Lim, L. R. Saunders, R. D. Stevens, C. B. Newgard, R. V. Farese, R. d. Cabo, S. Ulrich, K. Akassoglou, E. Verdin, Suppression of Oxidative Stress by β -Hydroxybutyrate, an Endogenous Histone Deacetylase Inhibitor, *Science* 339 (2013) 211-214.
- [8] Y. M. Han, T. Bedarida, Y. Ding, B. K. Somba, Q. Lu, Q. Wang, P. Song, M. H. Zou, β -Hydroxybutyrate Prevents Vascular Senescence through hnRNP A1-Mediated Upregulation of Oct4, *Mol Cell* 71 (2018) 1064-1078.e1065.

Reviewers' Comments:

Reviewer #2:

Remarks to the Author:

Authors have conducted additional studies and carefully revised the manuscript. There are no additional comments.

Reviewer #3:

Remarks to the Author:

The manuscript presents a novel method for single-cell metabolomics profiling. The new method can discriminate isomers and identify more metabolites in a single-cell level than other methods. Thus, it brings an interesting approach to the field. The manuscript has been partly rewritten to include more details and description that has improved the manuscript.

However, I still have three major concerns: Firstly, the authors show that MCF-7 with GHB/BHB isomer information can be divided into three clusters (Figure 5b). scRNA-seq data from MCF-7 cells also seem to form three clusters (Figure 6a). Next the authors want to show that gene expression and abundance values of three clusters from scRNA-seq and single-cell ID-organic cytoMS correlate. The evidence presented in the manuscript is, however, weak, as only two figures for two genes are shown (Figures 6c-d), with no correlation or statistical tests.

My second concern relates to the conclusion: "distribution heterogeneity of BHB signaling may endow cancer cells with variant ability to resist surrounding oxidative stress". This conclusion is based on the expression of single (cherry picked?) candidate genes, such as TCA cycle genes that are not even widely expressed in this data. Thus, the conclusion remains in the level of discussion rather than conclusion. The situation could be somewhat improved if an unbiased analysis of scRNA-seq data would have been shown to support the conclusion.

My third concern is about the methods section. The scRNA-seq analyses are still not properly made and described. They are also partly inconsistent, e.g. the methods section states that Wilcoxon Rank sum test was used, while the Figure legends state that t-test was used.

The language of the manuscript is now revised although some flaws remain, e.g. in writing numbers and verb tense, and especially the methods section still includes sections that are imprecise, poorly written, or even unintelligible.

L21-L22: Fine sub-typing of MCF-7 cells are first demonstrated by differential distribution of 3-hydroxybutanoic acid (BHB)

This sentence is unclear. If you write 'differential distribution' you should tell what you have compared, i.e. from what BHB differs?

L84-85: interpreting correlation between metabolites and biological process at the single-cell level.

This sentence should be revised. Do you mean to say relationship between metabolite abundance and gene expression at the single-cell level?

L192-193: KEGG annotation analysis revealed that these metabolites mainly cover basal energy metabolism, membrane transport and other cellular processes.

L198: d KEGG pathway annotation analysis from identified metabolites in 120 single cells.

If I read Fig 3d correctly, the figure shows frequency distribution of KEGG pathway annotation, not "KEGG annotation analysis". Please be careful when using a word 'analysis' because here it misleadingly refers to enrichment analysis.

L218-219: Fig. 4a-c shows the relative abundance ratio of the isomer pairs among 60 single cells

It would be clearer to the reader, if you would name the figures in the second line of Figure 4 as

4d-f. Also, the coloring of relative ratio figures is confusing because the same colors are used in the upper panels. Could you change the bar color of the lower panels and describe in the figure legend which way the intensity ratio has been calculated. Now it is not clear from the lower figure which isoform has higher intensity.

L241-L243: As shown in Fig. 5a, 4 types of tumors cells were successfully distinguished using t-distributed stochastic neighbor embedding (t-SNE) by using abundance of metabolites without isomers.

The language of this sentence should be revised.

Supplementary Fig. 12:

Legend of the color bar is missing. Exact definition of plotted values in the heatmap remain unclear.

L249-250: Relating bioinformatics algorithm also verified the clustering result (Supplementary Fig. 13).

Could you describe what is the 'relating bioinformatics algorithm'?

Supplementary Figure 13. Cluster number determination with metabolome data (with GHB/BHB) isomer using R. (a) Nbclust package and (b). within-cluster sum of squares (wss) test both verified that optimized cluster number is 3.

It is a good idea to study which cluster number would best describe the data. But it is still unclear which method was used to plot the Supplementary Figure 13a. For readers it is not informative to tell that you used R or Nbclust package rather you should name the method or the measure that you used and explain what the y-axis text 'Number of indices' refers to. It is also unclear why 5-6 clusters are missing from the figure. Please, see my comment on the "optimal number on clusters" below.

L251-253: The other two isomers, however, did not exhibit the ability to further distinguish between subtypes of MCF-7 cells (Supplementary Fig. 14).

This figure is a good adding, but you could still cluster these cells and show the difference in the relative ratio of isomers between clusters in the supplementary material. The methods section should also describe how UMAPs are made.

L255: It is common sense that...

Please revise = Pay attention to the use of academic language.

L265-266: Figure 5c: t test, two-sided

Instead of using three pairwise t-tests, linear model with contrasts should have been used. Description of the analysis is also missing from the methods section, as well as the description of how UMAPs in Figure 5b were constructed, did you use PCs, how many?

L271-273: The accuracy of the sequencing results was initially validated by expression verification of specific protein markers in MCF-7 cells (Supplementary Fig 16).

Add references to verify the choice of the 'specific protein markers'.

L301-302: Our results also indicated differential expression of hnRNP-A1 among cells (Fig. 6b and Supplementary Fig. 19).

Did you mean: "variability in hnRNP-A1 expression among cells" or "differential expression among clusters"?

L306-307: to further investigate correlation between BHB abundance and target protein expression.

Word 'relationship' instead of 'correlation' would be more precise here.

L309-312: As shown in Fig. 6c and 6d, cell population with higher expression abundance of MT2A/hnRNP A1 exhibited higher BHB relative abundance, which verified the positive correlation between BHB contents and target protein expression.

The conclusion is now based only on the figure showing slight increase in the BHB isomer relative

ratio with increasing fluorescence intensity. To convince the reader, you should add the actual correlation values.

Figure 6b:

Figure has two y-axis legends showing the same text but different scales.

L331-L334: Single-cell transcriptomic results show that expression of key enzymes covering TCA cycle and fatty acids synthesis remain relatively unchanged among all MCF-7 cells (Fig. 7b, 7d and Supplementary Fig. 22)

Figures 7b and 7c and Supplementary Figure 22 do not support this conclusion. As shown in the Supplementary Figure 22, percentage of expressed cells seem to be low in most of these genes, that's why the bar plots in Figures 7b and 7d are not very informative. i.e., the reason why expression of these genes remain unchanged is because most cells are not expressed. Thus, this conclusion is not valid. It is also unclear what the bar plots in Figures 7b and 7d represent, mean or median?

Supplementary Figure 19. UMAP plot of BHB downstream target proteins within MCF-7 cells

Feature plots are informative, but how did you generate them? From the color bar it looks like the expression values have been categorized and the actual normalized or scaled values are not shown.

Supplementary Figure 20. Expressed cell proportion of MT2A and hnRNP A1 among MCF-7 cells.

Bar plot of % expressed cells is a good adding but should show % of expressed cells in three clusters, because otherwise the biggest cluster dominates the results. From the feature plot it looks like in most genes % of expressed genes is lower in the right-most cluster.

From Revision report: "Similar to the previous inquiry, we performed statistical analysis using box plots to assess the expression of BHB-related target proteins, revealing significant variations in gene expression across distinct cell clusters. (see supplementary figure 21)"

Supplementary Figure 21 shows bar plots not box plots, although boxplots would have been more informative as they show the distribution of expression values not only mean (or median?) values. Now it is unclear what the bars represent. Also, plots could be ordered by genes rather than clusters. Note to rebuttal letter: boxplots are just visualizations not statistical tests.

L379-381: BHB downstream target proteins related to anti-oxidative stress, are positively expressed corresponding to BHB abundance.

What do you mean with "Positively expressed"? The sentence needs revision.

L381-382: Metabolites and key enzymes of TCA cycle and fatty acids synthesis remain unchanged in MCF-7 cells

The manuscript does not show evidence for this statement. See comment above.

Supplementary Figure 23 KEGG pathway enrichment bubble diagram. RNA degradation and neuron degenerative diseases related genes were significantly rich. Q-value (corrected P-value) ≤ 0.05 was used as the threshold of significantly enriched candidate genes.

It is unclear which gene set is significantly enriched, which gene sets has been compared and how the genes were chosen for the enrichment analysis. Without this knowledge it is impossible to interpret the figure. To improve the appearance of the figure, you could only show pathways with Q-value < 0.05 (or 0.1). From the current figure it is impossible to see which pathways are "significantly enriched". The description of the enrichment analysis is also missing from the methods section.

L517-L518: The optimal number of clusters was determined by Nbclust package and within-cluster sum of squares (wss) in R.

"Optimal" number of clusters is a tricky concept because it depends on the criteria used. It would

be better if you first describe which clustering method you used, how did you clustered the data, and then show that some measures support your choice to use three clusters. "number of clusters was determined by Nbclust package" is not a good description of what has been done. Rather you should tell the which method or measure was used.

L550-562

This section is not understandable due to poor English writing, please revise.

L550:552: RNA reads were be aligned using STAR alignment software. Raw reads in the.fastq files of MCF7 cells were processed in the Cell Ranger Software Suite (10x Genomics Cell Ranger 6.1.2), and unique molecular identifier (UMI) matrices were generated.

This section should be revised. The text is like written in the wrong order: Normally Cell Ranger is used to process the data to generate fastq files and UMI counts.

L553-L555: Briefly, filter out the cells with the number of genes identified less than 200 or greater than the maximum number of genes 90%.

Do you mean you filtered out cells if the number of expressed genes was above 90% percentile? Please revise.

L557-558: Variable features selection was conducted using the function of FindVariableFeatures 2000 in Seurat, Doublet detection.

Did you apply doublet detection?

L558-559: 2000 genes with the highest degree of variation are selected for downstream dimensionality reduction and clustering.

Description of how dimensional reduction and clustering was made is still missing from the methods section.

L560-562: Marker genes were selected by Software Seurat (3.0.2), Parameters were set as follows: findallmarkers: only.pos = TRUE; min.pct = 0.1; test.use = wilcox; logfc.threshold = 0.25, p.adjust = 0.05.

Did you mean that you selected marker genes for clusters? Did you run differential expression analysis between clusters? Here you state that you used Wilcoxon rank sum test but in Figures you state that you used t-test, which is very confusing. For differential expression (DE) analysis between clusters, t-test is not even a correct test. Instead, you could use for example Deseq2. Furthermore, the summary of the full DE analysis is still missing.

L568-570: Single cell sequencing data that support the findings of this study have been deposited in Genome Sequence Archive (GSA) with the accession codes PRJCA018042. Source data are provided with this paper.

As usual, single cell gene expression data should be fully submitted to GEO and made it open to reviewers.

Supplementary Table 1. Marker gene list of single cell clusters, which covers DNA and RNA binding as well as transcription. (log(FC). threshold = 0.25; Adjust P value.= 0.05)

Revise the text in the parenthesis.

Reviewer #4:

Remarks to the Author:

Dear Editor,

per your request I have read the manuscript and the rebuttal of the comments to reviewer 1. Although i have not seen the first submitted version, from reading the revised manuscript, i get the impression that the authors have made significant changes to the manuscript after the initial round of reviews. Reading the new version and the rebuttal i think they have sufficiently addressed the comments of reviewer 1.

With the broad audience of nature communications in mind, i do want to raise two points.

1. reviewer 1 comment 10 already pointed out the following: "Line 62: "online" -- given the varied readership of the journal, it may be worth clarifying here what online vs offline methods are explicitly"

the authors have indeed aimed to address this by giving examples of online and offline methods but have not clarified the definition of an online/offline method, which is not common knowledge for the general reader.

2. The overall writing of some sentences in especially the introduction can be improved. examples of this are lines 44 and 51.

Dear Respected Editor and Reviewers,

Thank you very much for your reviewing and handling our manuscript entitled “In-depth organic mass cytometry reveals differential contents of 3-hydroxybutanoic acid on single cell level” (No. NCOMMS-23-27831B). We gratefully acknowledge all reviewers for their evaluation and valuable comments for further revising and improving our manuscript.

The point-to-point responses have been carefully prepared to response reviewers’ comments, and the manuscript was revised following the reviews’ comments, especially reviewer 3’s comments on single cell sequencing, and to emphasize the innovations of our work in a clearer way. The language of the manuscript has been professionally edited through SPRINGER NATURE Author Services system. All the changes were highlighted in yellow in the revised manuscript. The point-to-point responses to the reviewers’ comments are listing below.

Reviewer #3

1. *The authors show that MCF-7 with GHB/BHB isomer information can be divided into three clusters (Figure 5b). scRNA-seq data from MCF-7 cells also seem to form three clusters (Figure 6a). Next the authors want to show that gene expression and abundance values of three clusters from scRNA-seq and single-cell ID-organic cytoMS correlate. The evidence presented in the manuscript is, however, weak, as only two figures for two genes are shown (Figures 6c-d), with no correlation or statistical tests.*

Response: Thank you very much for your questions and comments. Following your comments, we performed an unbiased analysis of differentially expressed genes starting from the single cell sequencing data. Volcano plot of differential genes expression analysis and a list of differential genes between different cell clusters were shown in Supplementary Figure 19 and Supplementary Dataset 2 in revised manuscript. Our differential analysis results confirmed the genes encoding downstream transcriptional regulation proteins of BHB, including *MT2A*, *hnRNPA1*, as well as *GATA3*, *SULT2B1*, and *SOD1* etc. that exhibited differential expression among cell clusters. They suggest potential variations in BHB abundance within distinct cell clusters in MCF-7 cells. To establish a connection between single-cell metabolomics and sequencing results, we took *MT2A* and *hnRNPA1*, the two BHB downstream target proteins with most significant difference, as examples and conducted FACS validation experiments.

Due to technical limitations, establishing a one-to-one correspondence between cells in the two experiments is not feasible. Therefore, we analyze the FACS data by utilizing average fluorescence intensity. (see revised Figure 6). We observed that increased expression of MT2A/hnRNPA1 genes correlated with a gradual increase of BHB content, indicating a linear correlation between them (Pearson's $r = 0.99$). Therefore, both transcriptomic and metabolomics analyses confirm the differential levels of BHB content in MCF-7 cells.

2. *My second concern relates to the conclusion: “distribution heterogeneity of BHB signaling may endow cancer cells with variant ability to resist surrounding oxidative stress”. This conclusion is based on the expression of single (cherry picked?) candidate genes, such as TCA cycle genes that are not even widely expressed in this data. Thus, the conclusion remains in the level of discussion rather than conclusion. The situation could be somewhat improved if an unbiased analysis of scRNA-seq data would have been shown to support the conclusion.*

Response: Thank you very much for your comments and suggestions. Considering our previous limited and improper single-cell transcriptomic analysis, after discussion with our collaborator and re-analyzing those data together, we moved this part to the supplementary data. Importantly, following your valuable suggestion, we performed an unbiased analysis of differentially expressed genes starting from the single cell sequencing data. Volcano plot of differential genes expression analysis and a list of differential genes between cell clusters were shown in revised Supplementary Figure 19 and Supplementary Dataset 2. The differential analysis results confirmed the downstream anti-oxidative stress relating genes, such as *MT2A* and *SOD1*, exhibited differential expression among cell clusters. The KEGG enrichment analysis also indicate significant enrichment of pathways related to oxidative phosphorylation (see revised Supplementary Figure 25). In addition, no positive correlation between the abundance of metabolites associated with TCA cycle and fatty acid metabolism with BHB contents (see revised Supplementary Figure 24) were found, suggesting that these metabolic pathways are not related with BHB content or cell subtype clustering. Based on all above results, we have carefully re-organized the related section in the main text (see page 15 line 341-351).

3. *My third concern is about the methods section. The scRNA-seq analyses are still not properly made and described. They are also partly inconsistent, e.g. the methods section states that Wilcoxon Rank sum test was used, while the Figure legends state that t-test was used.*

Response: Thank you very much for your valuable suggestions. We have carefully revised this section with the help of two collaborators with sequencing background (see page 23 line 560-581). Furthermore, due to the non-normal distribution of our single-cell samples in this study, t-tests are not applicable. Consequently, we have adapted our approach to comparing differences in all single-cell data by employing Wilcoxon rank-sum tests (see revised Supplementary Figure 19) and Kruskal-Wallis test. (see revised Supplementary Figure 20)

4. *The language of the manuscript is now revised although some flaws remain, e.g. in writing numbers and verb tense, and especially the methods section still includes sections that are imprecise, poorly written, or even unintelligible.*

Response: Thank you very much for your comments. The language of the whole manuscript has been edited using SPRINGER NATURE Author Services system and the editing certificate has been uploaded.

5. *L21-L22: Fine sub-typing of MCF-7 cells are first demonstrated by differential distribution of 3-hydroxybutanoic acid (BHB).*

--This sentence is unclear. If you write 'differential distribution' you should tell what you have compared, i.e. from what BHB differs?

Response: Thank you for your comments. Sorry for our unclear writing, following your advice, we have refined it as "Fine subtyping of MCF-7 cells was first demonstrated by an investigation on the differential levels of 3-hydroxybutanoic acid among clusters." (see page 2 line 25-27).

6. *L84-85: interpreting correlation between metabolites and biological process at the single-cell level.*

--This sentence should be revised. Do you mean to say relationship between metabolite abundance and gene expression at the single-cell level?

Response: Thank you for your comments. We have revised it as "This work proposes and validates an approach for high-throughput and in-depth single-cell metabolomics, which paves the way for comprehensive clarifying metabolic heterogeneity in cells and interpreting complicated biological processes at the single-cell level" (see page 5 line 92-95).

7. *L192-193: KEGG annotation analysis revealed that these metabolites mainly cover basal energy metabolism, membrane transport and other cellular processes.*

L198: d KEGG pathway annotation analysis from identified metabolites in 120 single cells.

--If I read Fig 3d correctly, the figure shows frequency distribution of KEGG pathway annotation, not “KEGG annotation analysis”. Please be careful when using a word ‘analysis’ because here it misleadingly refers to enrichment analysis.

Response: Thank you for your valuable suggestions. We have made a clear distinction between KEGG annotation and KEGG enrichment analysis in the main text, for example, KEGG annotation in revised Figure 3. and KEGG enrichment analysis in revised Supplementary Figure 25.

8. *L218-219: Fig. 4a-c shows the relative abundance ratio of the isomer pairs among 60 single cells.*

--It would be clearer to the reader, if you would name the figures in the second line of Figure 4 as 4d-f. Also, the coloring of relative ratio figures is confusing because the same colors are used in the upper panels. Could you change the bar color of the lower panels and describe in the figure legend which way the intensity ratio has been calculated. Now it is not clear from the lower figure which isoform has higher intensity.

Response: Thank you for your valuable comments. Following your suggestion, we have implemented numbering for Figure 4 and made revisions to the color scheme in the lower panels. Furthermore, we have included comprehensive explanations regarding the calculation method used for determining isomer relative ratios. (see page 11 line 240-242)

9. *L241-L243: As shown in Fig. 5a, 4 types of tumors cells were successfully distinguished using t-distributed stochastic neighbor embedding (t-SNE) by using abundance of metabolites without isomers.*

--The language of this sentence should be revised.

Response: Thank you for your suggestion. We have rewritten this sentence into “four types of tumor cells were successfully distinguished by the abundance of metabolites without isomers by t-distributed stochastic neighbor embedding (t-SNE)”. (see page 11 line 248-250)

10. *Supplementary Fig. 12: Legend of the color bar is missing. Exact definition of plotted values in the heatmap remain unclear.*

Response: Thank you for your valuable suggestion. Following your suggestion, we have supplemented the legend and clarified the exact definition of plotted values (see revised Supplementary Figure 12)

11. L249-250: *Relating bioinformatics algorithm also verified the clustering result (Supplementary Fig. 13).*

--Could you describe what is the 'relating bioinformatics algorithm'?

Response: Thank you for your suggestion. We have corrected it as “The K-means clustering method in the Nbclust package of R was utilized to verify the clustering results”. (see page 12 line 256-257).

12. *Supplementary Figure 13. Cluster number determination with metabolome data (with GHB/BHB) isomer using R. (a) Nbclust package and (b). within-cluster sum of squares (wss) test both verified that optimized cluster number is 3.*

--It is a good idea to study which cluster number would best describe the data. But it is still unclear which method was used to plot the Supplementary Figure 13a. For readers it is not informative to tell that you used R or Nbclust package rather you should name the method or the measure that you used and explain what the y-axis text 'Number of indices' refers to. It is also unclear why 5-6 clusters are missing from the figure. Please, see my comment on the "optimal number on clusters" below.

Response: Thank you for your suggestion and sorry for our carelessness. We have revised the whole section and added detailed parameters of the Nbclust package in method section. The y-axis text 'Number of indices' was added and explained (see page 24 line 584-589). Among all 30 indices, no indices proposed 5 or 6 as the best number of clusters, that's why 5-6 clusters are missing from the figure. Detailed indices definition and corresponding calculation methods refers to the literature^[1].

13. L251-253: *The other two isomers, however, did not exhibit the ability to further distinguish between subtypes of MCF-7 cells (Supplementary Fig. 14).*

--This figure is a good adding, but you could still cluster these cells and show the difference in the relative ratio of isomers between clusters in the supplementary material. The methods section should also describe how UMAPs are made.

Response: Thank you for your valuable suggestion. We have clustered these cells into three clusters using K-mean method, and supplemented the boxplots to show the difference in the relative ratio of the other two isomers between clusters (see revised Supplementary Fig. 15). There was no significant difference among clusters. Detailed UMAP reduction and clustering

have been described in the method section. (see page 24 line 590-596)

14. L255: *It is common sense that...*

--Please revise = Pay attention to the use of academic language.

Response: Thank you for your suggestion. We have revised and corrected it as “Notably, MCF-7 cells are derived from breast tumors and are dominated by the luminal A subtype”. (see page 12 line 269-270)

15. L265-266: *Figure 5c: t test, two-sided*

--Instead of using three pairwise t-tests, linear model with contrasts should have been used. Description of the analysis is also missing from the methods section, as well as the description of how UMAPs in Figure 5b were constructed, did you use PCs, how many?

Response: Thank you for your valuable suggestion. Following your suggestion, we have revised this part. The Kruskal-Wallis test was employed to perform statistical analysis on differences among multiple independent single cell samples, with detailed statistical methods provided in legend of Figure 5c and 6c. Additionally, specific parameter settings for UMAP dimensional reduction can be found in the same section (see page 24 line 590-596). The UMAP method parameters are the same as the default parameters in the UMAP package in R.

16. L271-273: *The accuracy of the sequencing results was initially validated by expression verification of specific protein markers in MCF-7 cells (Supplementary Fig 16).*

--Add references to verify the choice of the ‘specific protein markers’.

Response: Thank you for your valuable suggestion. Following your suggestion, we have supplemented more references to verify the choice of “specific protein markers” (see revised Supplementary Fig. 18).

17. L301-302: *Our results also indicated differential expression of hnRNP-A1 among cells (Fig. 6b and Supplementary Fig. 19).*

--Did you mean: “variability in hnRNP-A1 expression among cells” or “differential expression among clusters”?

Response: Thank you for your valuable questions. “differential expression among clusters” is the meaning that we want to express. We have made the correction and modification in the main text. (see page 14 line 313-314)

18. L306-307: *to further investigate correlation between BHB abundance and target protein*

expression.

--Word 'relationship' instead of 'correlation' would be more precise here.

Response: Thank you for your valuable suggestions. Following your suggestion, we have revised and corrected all related part. (see page 14 line 318 and 323, page 5 line 91, page 2 line 30)

19. L309-312: *As shown in Fig. 6c and 6d, cell population with higher expression abundance of MT2A/hnRNP A1 exhibited higher BHB relative abundance, which verified the positive correlation between BHB contents and target protein expression.*

--The conclusion is now based only on the figure showing slight increase in the BHB isomer relative ratio with increasing fluorescence intensity. To convince the reader, you should add the actual correlation values.

Response: Thank you for your valuable suggestions. Following your suggestion, we had supplemented the Pearson correlation values in the revised Figure 6.

20. Figure 6b: *Figure has two y-axis legends showing the same text but different scales.*

Response: Thank you for your valuable suggestions. We have redrawn Figure 6b by using 2 figures to make it much clear.

21. L331-L334: *Single-cell transcriptomic results show that expression of key enzymes covering TCA cycle and fatty acids synthesis remain relatively unchanged among all MCF-7 cells (Fig. 7b, 7d and Supplementary Fig. 22)*

--Figures 7b and 7c and Supplementary Figure 22 do not support this conclusion. As shown in the Supplementary Figure 22, percentage of expressed cells seem to be low in most of these genes, that's why the bar plots in Figures 7b and 7d are not very informative. i.e., the reason why expression of these genes remain unchanged is because most cells are not expressed. Thus, this conclusion is not valid. It is also unclear what the bar plots in Figures 7b and 7d represent, mean or median?

Response: Thank you for your valuable suggestions. Considering our previous limited and improper single-cell transcriptomic analysis, after discussion with our collaborator and reanalyzing those data, we moved this part to the supplementary data and re-organized this part in the manuscript. We now found that the levels of metabolites associated with the tricarboxylic acid (TCA) cycle and fatty acid metabolism remain relatively stable across all three cell clusters,

indicating no significant variations in metabolic activity along these pathways between the cell clusters. Bar plots of Figures 7b and 7d in previous manuscript represent mean, overall considering and re-evaluating the content of this work, we have deleted Figure 7 in the revised manuscript.

22. *Supplementary Figure 19. UMAP plot of BHB downstream target proteins within MCF-7 cells*

--Feature plots are informative, but how did you generate them? From the color bar it looks like the expression values have been categorized and the actual normalized or scaled values are not shown.

Response: Thank you very much for your questions. We performed log₂ normalization on the expression intensity and displayed the global expression pattern of specific genes across the entire sample through color changes. (see revised Supplementary Figure 22)

23. *Supplementary Figure 20. Expressed cell proportion of MT2A and hnRNP A1 among MCF-7 cells.*

--Bar plot of % expressed cells is a good adding but should show % of expressed cells in three clusters, because otherwise the biggest cluster dominates the results. From the feature plot it looks like in most genes % of expressed genes is lower in the right-most cluster.

Response: Thank you very much for your questions. We have made corresponding revision and exhibited expressed cell proportion of MT2A and hnRNPA1 among clusters. Low expression levels of MT2A and hnRNPA1 in cluster 1 may resulted in a lower proportion of expressing cells. (see revised Supplementary Figure 23)

24. *From Revision report: "Similar to the previous inquiry, we performed statistical analysis using box plots to assess the expression of BHB-related target proteins, revealing significant variations in gene expression across distinct cell clusters. (see supplementary figure 21)"*

--Supplementary Figure 21 shows bar plots not box plots, although boxplots would have been more informative as they show the distribution of expression values not only mean (or median?) values. Now it is unclear what the bars represent. Also, plots could be ordered by genes rather than clusters.

Response: Thank you very much for your suggestions. Following your suggestion, we did the unbiased differential expression gene analysis. The analysis results revealed the significant difference of BHB-related target genes. Therefore, we replaced the bar plots with a volcano plot

of differentially expressed genes. (see revised Supplementary Figure 19)

25. L379-381: *BHB downstream target proteins related to anti-oxidative stress, are positively expressed corresponding to BHB abundance.*

--What do you mean with "Positively expressed"? The sentence needs revision.

Response: Thank you very much for your suggestions. We had rewritten it as "The expression levels of MT2A and hnRNP A1, genes encoding downstream target proteins associated with the antioxidant stress response, exhibit a positive relationship with the abundance of BHB". (see page 17 line 383-385).

26. L381-382: *Metabolites and key enzymes of TCA cycle and fatty acids synthesis remain unchanged in MCF-7 cells*

--The manuscript does not show evidence for this statement. See comment above.

Response: Thank you very much for your suggestions. Following your suggestion, we had made corresponding revision as shown in question 21.

27. *Supplementary Figure 23 KEGG pathway enrichment bubble diagram. RNA degradation and neuron degenerative diseases related genes were significantly rich. Q-value (corrected P-value) ≤ 0.05 was used as the threshold of significantly enriched candidate genes.*

--It is unclear which gene set is significantly enriched, which gene sets has been compared and how the genes were chosen for the enrichment analysis. Without this knowledge it is impossible to interpret the figure. To improve the appearance of the figure, you could only show pathways with $Q\text{-value} < 0.05$ (or 0.1). From the current figure it is impossible to see which pathways are "significantly enriched". The description of the enrichment analysis is also missing from the methods section.

Response: Thank you very much for your valuable questions. Following your question, we have carefully revised this section. Differently expressed gene across different cell clusters were identified using the FindMarkers function in Seurat with parameters 'logfc. Threshold >0.25, minPct>0.1 and Padj \leq 0.05'. KEGG enrichment analysis displayed the significantly enriched pathways among clusters based on differentially expressed genes. The pathways that showed significant enrichment are highlighted in red background, while the specific methods employed for conducting the KEGG enrichment analysis can be found in the methods section (see page 24 line 578-581).

28. L517-L518: *The optimal number of clusters was determined by Nbclust package and within-cluster sum of squares (wss) in R.*

-- *“Optimal” number of clusters is a tricky concept because it depends on the criteria used. It would be better if you first describe which clustering method you used, how did you clustered the data, and then show that some measures support your choice to use three clusters. “number of clusters was determined by Nbclust package” is not a good description of what has been done. Rather you should tell the which method or measure was used.*

Response: Thank you very much for your suggestion. Following your suggestion, we have rewritten this part and added detailed method parameters of the Nbclust package in method section (see page 24 line 584-589).

29. L550-562: *This section is not understandable due to poor English writing, please revise.*

Response: Thank you very much for your valuable suggestion. We have carefully rewritten the whole method section of single-cell sequencing experiments, which can be found in the page 23 line 560-581.

30. L550:552: *RNA reads were be aligned using STAR alignment software. Raw reads in the.fastq files of MCF7 cells were processed in the Cell Ranger Software Suite (10x Genomics Cell Ranger 6.1.2), and unique molecular identifier (UMI) matrices were generated.*

--*This section should be revised. The text is like written in the wrong order: Normally Cell Ranger is used to process the data to generate fastq files and UMI counts.*

Response: Thank you very much for your valuable questions. This section has been carefully revised in the page 23 line 560-564 and the text has been rewritten into “Cell Ranger Software Suite (10x Genomics Cell Ranger 6.1.2) is used to process the raw data to generate fastq files and unique molecular identifier (UMI) matrices of MCF7 cells”.

31. L553-L555: *Briefly, filter out the cells with the number of genes identified less than 200 or greater than the maximum number of genes 90%.*

--*Do you mean you filtered out cells if the number of expressed genes was above 90% percentile? Please revise.*

Response: Thank you very much for your valuable suggestion. We have made revisions to the methods section of single-cell sequencing experiments, which can be found in the page 23 line 567-572. In principle, cells with less than 200 detected genes or cells with >90% of the

proportion of the maximum genes were filtered out. The latter aims to avoid potential doublets or cells that highly expressed mitochondrial genes.

32. L557-558: *Variable features selection was conducted using the function of FindVariableFeatures 2000 in Seurat, Doubletdetection.*

--Did you apply doublet detection?

Response: Thank you very much for your valuable suggestion. We have made revisions to the methods section of single-cell sequencing experiments, which can be found in the page 23 line 571. In detail, potential doublets were identified and removed by DoubletDetection function.

33. L558-559: *2000 genes with the highest degree of variation are selected for downstream dimensionality reduction and clustering.*

--Description of how dimensional reduction and clustering was made is still missing from the methods section.

Response: Thank you very much for your valuable suggestion. We have added the description of dimensional reduction and clustering in the method section, which can be found in the page 24 line 584-596. In detail, principal component (n=15) analysis was conducted using only the 2000 highly variable genes in the dataset. UMAP was then used for two-dimensional visualization of the resulting clusters.

34. L560-562: *Marker genes were selected by Software Seurat (3.0.2), Parameters were set as follows: findallmarkers: only.pos = TRUE; min.pct = 0.1; test.use = wilcox; logfc.threshold = 0.25, p.adjust = 0.05.*

--Did you mean that you selected marker genes for clusters? Did you run differential expression analysis between clusters? Here you state that you used Wilcoxon rank sum test but in Figures you state that you used t-test, which is very confusing. For differential expression (DE) analysis between clusters, t-test is not even a correct test. Instead, you could use for example Deseq2. Furthermore, the summary of the full DE analysis is still missing.

Response: Thank you very much for your valuable questions and sorry for our carelessness. Actually, we didn't select marker genes for clusters. Following your suggestion, we performed an unbiased analysis of differentially expressed genes starting from the single cell sequencing data. Wilcoxon rank sum test was employed to assess statistical significance. Volcano plot of differential genes expression analysis and a list of differential genes between different cell

clusters were shown in revised Supplementary Figure 19 and Supplementary Dataset 2.

35. *L568-570: Single cell sequencing data that support the findings of this study have been deposited in Genome Sequence Archive (GSA) with the accession codes PRJCA018042. Source data are provided with this paper.*

--As usual, single cell gene expression data should be fully submitted to GEO and made it open to reviewers.

Response: Thank you very much for your valuable suggestion. We have fully submitted single cell gene expression data of MCF-7 cells to GEO database and made it open to the public.

36. *Supplementary Table 1. Marker gene list of single cell clusters, which covers DNA and RNA binding as well as transcription. ($\log(FC)$, threshold = 0.25; Adjust P value. = 0.05)*

--Revise the text in the parenthesis.

Response: Thank you very much for your valuable question and sorry for our carelessness. Since we didn't perform the marker gene analysis, we have removed marker gene analysis contents and supplemented with description of differentially expressed genes analysis. (see revised Supplementary Figure 19)

Reviewer #4

1. *reviewer 1 comment 10 already pointed out the following: "Line 62: "online" -- given the varied e readership of the journal, it may be worth clarifying here what online vs offline methods are explicitly", the authors have indeed aimed to address this by giving examples of online and offline methods but have not clarified the definition of an online/offline method, which is not common knowledge for the general reader.*

Response: Thank you very much for your valuable and significant feedback. We have supplemented the definition of online/offline single cell metabolomics methods. (see page 3 line 47-51)

2. *The overall writing of some sentences in especially the introduction can be improved. examples of this are lines 44 and 51.*

Response: Thank you very much for your suggestions. The language of the whole article has been polished by Language Editing Service system of SPRINGER NATURE (certificate attached).

References:

- [1] M. Charrad, N. Ghazzali, V. Boiteau, A. Niknafs, NbClust: An R Package for Determining the Relevant Number of Clusters in a Data Set, *Journal of Statistical Software*, 2014, 61, 1-36.

Reviewers' Comments:

Reviewer #3:

Remarks to the Author:

The authors have addressed all my concerns and requests in the revised manuscript. I have only a few minor comments to make.

• **L226-227: Fig. 4a-c shows the relative abundance ratio of the isomer pairs among 60 single cells.**

Please update the main text to reflect the changes in figure numbering. The sentence should now refer to Figs 4d-f."

• **L254-L257 Conversely, after the BHB/GHB isomer information was combined, 3 clusters of cells were distinguished through uniform manifold approximation and projection (UMAP). The K-means clustering method in the Nbclust package of R was utilized to verify the clustering results (Supplementary Fig. 13).**

Kindly refer to the explanations provided by both myself and the other reviewer regarding why data clustering with UMAP is not appropriate. It appears that you reverted the text to a version similar to the initial manuscript, which was incorrect.

• In general, it suffices to state in the results section:

Conversely, after the BHB/GHB isomer information was combined, 3 clusters of cells were distinguished (Fig 5b, Supplementary Fig. 13).

Readers can find the precise clustering procedure, software used, and an explanation of how the number of clusters was determined in the methods section.

• **Figure 5. Clustering of MCF-7 cells with GHB/BHB isomer information. a t-SNE visualization of 4 types of tumor cells. b UMAP visualization of MCF-7 cells with and without BHB/GHB isomers. (n=138 cells). c Relative abundance of GHB isomers (MS peak intensity at m/z 59.0138 vs. m/z 57.0345) among 3 clusters (n=138 cells in total). L257-258: By using the relative ratio of BHB/GHB isomers from 3 clusters**

Please maintain consistency and use BHB/GHB consistently throughout both the text and figures. Currently, Figure 5c's legend indicates a comparison of GHB abundance, while the text refers to the relative ratio of BHB/GHB isomers. Additionally, the figure displays GHB/BHB information, whereas the text mentions BHB/GHB isomer information.

• **278-280: The average number of detected genes and transcripts (UMI counts) per cell were 1897 and 7072, respectively. (Supplementary Table. 1 and Supplementary Fig 17).**

This sentence does not add relevant information to the results and may be better suited for the methods section.

• **Supplementary Figure 19. Volcano plot of differential genes expression analysis among 3 clusters from single cell sequencing data. (n=10373 cells in total, dots represent differential genes, with up-regulated genes marked as red and down-regulated genes marked as blue. BHB downstream target proteins were annotated (wilcoxon test, $\log_2|FC| \geq 0.25$, Q values ≤ 0.05).**

I was wondering why you use the threshold of $\log_2|FC| \geq 0.25$ as the level of significant genes. This threshold appears to result in many genes with only small difference between clusters.

• **Figure 6b Box plot of the normalized expression levels of MT2A (left orange scale) and hnRNP A1 (right blue scale) among the clusters. (n=10373 cells in total). (Kruskal-Wallis test, *** denoted p values < 0.001)**

Revise the the text in the parahthesis according to the changes in the figure (left orange scale) -> (upper) and (right blue scale) -> (lower)

• **L572-573 Cell cycle analysis was performed by using the CellCycleScoring function in**

the Seurat program.

The paper does not present results from the Cell cycle analysis. If these results are not presented, you can omit this sentence from the Methods section.

• L575-576: U-MAP was then used to visualize the resulting clusters in two-dimension. Differentially expressed genes across different cell clusters were...

In this sentence, you explain how you visualized clustering and used it in the DE analyses, but omitted the description of how clustering was performed. While clustering is described for single-cell metabolome data, it is not addressed for scRNA-seq.

• L580 the resulted P-values was corrected to Q values by BH method.

Please write the acronym BH in full.

• Review response (33): In detail, principal component (n=15) analysis was conducted using only the 2000 highly variable genes in the dataset. UMAP was then used for two-dimensional visualization of the resulting clusters.

This detail could be added to the methods section. Currently, the method section does not indicate that 15 principal components were used to perform UMAP, if indeed that was the case.

Dear Respected Editor and Reviewers,

Thank you very much for your reviewing and handling our manuscript entitled “In-depth organic mass cytometry reveals differential contents of 3-hydroxybutanoic acid on single cell level” (No. NCOMMS-23-27831B). We gratefully acknowledge all reviewers for their evaluation and valuable comments for further revising and improving our manuscript.

The point-to-point responses have been carefully prepared to response reviewer 3’s comments, and the manuscript was revised following the reviewer 3’s comments mainly on description of single cell sequencing method. All the changes were highlighted in yellow in the revised manuscript.

The point-to-point responses to the reviewers’ comments are listing below.

Reviewer #3

1. *L226-227: Fig. 4a-c shows the relative abundance ratio of the isomer pairs among 60 single cells.*

—Please update the main text to reflect the changes in figure numbering. The sentence should now refer to Figs 4d-f.

Response: Thank you for your suggestion and sorry for our carelessness. We have corrected the sentence into “Fig. 4d-f shows the relative abundance ratio of the isomer pairs among 60 single cells” in the revised manuscript. (see page 10 line 228-229)

2. *L254-L257 Conversely, after the BHB/GHB isomer information was combined, 3 clusters of cells were distinguished through uniform manifold approximation and projection (UMAP). The K-means clustering method in the Nbclust package of R was utilized to verify the clustering results (Supplementary Fig. 13).*

—Kindly refer to the explanations provided by both myself and the other reviewer regarding why data clustering with UMAP is not appropriate. It appears that you reverted the text to a version similar to the initial manuscript, which was incorrect. In general, it suffices to state in the results section: “Conversely, after the BHB/GHB isomer information was combined, 3 clusters of cells were distinguished (Fig 5b, Supplementary Fig. 13).”

Response: Thank you very much for your comments and suggestions. We re-referred to the all-

reviewer's comments regarding why data clustering with UMAP is not appropriate. Following your suggestions, We have revised it as "Conversely, after the BHB/GHB isomer information was combined, 3 clusters of cells were distinguished (Fig 5b, Supplementary Fig. 13)." (see page 11 line 257-258)

3. *Figure 5. Clustering of MCF-7 cells with GHB/BHB isomer information. a t-SNE visualization of 4 types of tumor cells. b UMAP visualization of MCF-7 cells with and without BHB/GHB isomers. (n=138 cells). c Relative abundance of GHB isomers (MS peak intensity at m/z 59.0138 vs. m/z 57.0345) among 3 clusters (n=138 cells in total). L257-258: By using the relative ratio of BHB/GHB isomers from 3 clusters.*

--Please maintain consistency and use BHB/GHB consistently throughout both the text and figures. Currently, Figure 5c's legend indicates a comparison of GHB abundance, while the text refers to the relative ratio of BHB/GHB isomers. Additionally, the figure displays GHB/BHB information, whereas the text mentions BHB/GHB isomer information.

Response: Thank you for your suggestion and sorry for our carelessness. We have uniformed all the descriptive order on BHB/GHB in the figures and text of revised manuscript. (see page 12 line 264-270, page 13 line 280).

4. *L278-280: The average number of detected genes and transcripts (UMI counts) per cell were 1897 and 7072, respectively. (Supplementary Table. 1 and Supplementary Fig 17).*

--This sentence does not add relevant information to the results and may be better suited for the methods section.

Response: Thank you very much for your valuable suggestions. We have transferred this sentence to the methods section (see page 23 line 567-569).

5. *Supplementary Figure 19. Volcano plot of differential genes expression analysis among 3 clusters from single cell sequencing data. (n=10373 cells in total, dots represent differential genes, with up-regulated genes marked as red and down-regulated genes marked as blue. BHB downstream target proteins were annotated (wilcoxon test, $\log_2|FC| \geq 0.25$, Q values ≤ 0.05).*

--I was wondering why you use the threshold of $\log_2|FC| \geq 0.25$ as the level of significant genes. This threshold appears to result in many genes with only small difference between clusters.

Response: Thank you very much for your comments. The default value of $\log_2|FC|$ in the findmarkers function of seurat package is set to 0.1. To ensure the significance of differential genes, we have retained the default value of 0.25 for this parameter in the BGI data processing platform. In fact, many literatures^[1-3] on single-cell sequencing also set the threshold at 0.25, so we did not adjust this parameter any more.

6. *Figure 6b Box plot of the normalized expression levels of MT2A (left orange scale) and hnRNP A1 (right blue scale) among the clusters. (n=10373 cells in total). (Kruskal-Wallis test, *** denoted p values < 0.001)*

--Revise the the text in the parathesis according to the changes in the figure (left orange scale)
-> (upper) and (right blue scale) -> (lower)

Response: Thank you for your suggestion and sorry for our carelessness. We have corrected the legend of figure 6b into “Box plot of the normalized expression levels of MT2A (upper) and hnRNP A1 (lower) among the clusters. (n=10373 cells in total). (Kruskal-Wallis test, *** denoted p values < 0.001)” (see page 15 line 333-335).

7. *L572-573 Cell cycle analysis was performed by using the CellCycleScoring function in the Seurat program.*

--The paper does not present results from the Cell cycle analysis. If these results are not presented, you can omit this sentence from the Methods section.

Response: Thank you for the reminder. We have omitted relevant contents about cell cycle analysis in the methods section of revised manuscript.

8. *L575-576: U-MAP was then used to visualize the resulting clusters in two-dimension. Differentially expressed genes across different cell clusters were...*

--In this sentence, you explain how you visualized clustering and used it in the DE analyses, but omitted the description of how clustering was performed. While clustering is described for single-cell metabolome data, it is not addressed for scRNA-seq.

Response: Thank you for your valuable suggestions. We have supplemented detailed description of UMAP clustering process in the method section (see page 24 line 594-596).

9. *L580 the resulted P-values was corrected to Q values by BH method.*

--Please write the acronym BH in full.

Response: Thank you for your valuable comments and sorry for our carelessness. We have

replaced “BH” with “Benjamini-Hochberg method” in the method section. (see page 24 line 581)

10. *Review response (33): In detail, principal component (n=15) analysis was conducted using only the 2000 highly variable genes in the dataset. UMAP was then used for two-dimensional visualization of the resulting clusters.*

--This detail could be added to the methods section. Currently, the method section does not indicate that 15 principal components were used to perform UMAP, if indeed that was the case.

Response: Thank you for your suggestions. We have added the sentence to the methods section (see page 23 line 574-576), and detailed UMAP algorithm parameters were also supplemented in the revised manuscript (see page 24 line 594-596).

References:

- [1] H. Sun, L. Zhang, Z. Wang, D. Gu, M. Zhu, Y. Cai, L. Li, J. Tang, B. Huang, B. Bosco, N. Li, L. Wu, W. Wu, L. Li, Y. Liang, L. Luo, Q. Liu, Y. Zhu, J. Sun, L. Shi, T. Xia, C. Yang, Q. Xu, X. Han, W. Zhang, J. Liu, D. Meng, H. Shao, X. Zheng, S. Li, H. Pan, J. Ke, W. Jiang, X. Zhang, X. Han, J. Chu, H. An, J. Ge, C. Pan, X. Wang, K. Li, Q. Wang, Q. Ding, Single-cell transcriptome analysis indicates fatty acid metabolism-mediated metastasis and immunosuppression in male breast cancer, *Nat. Commun.* 14 (2023) 5590-5590.
- [2] G. Gambardella, G. Viscido, B. Tumaini, A. Isacchi, R. Bosotti, D. di Bernardo, A single-cell analysis of breast cancer cell lines to study tumour heterogeneity and drug response, *Nat. Commun.* 13 (2022) 1714.
- [3] M. VanInsberghe, J. van den Berg, A. Andersson-Rolf, H. Clevers, A. van Oudenaarden, Single-cell Ribo-seq reveals cell cycle-dependent translational pausing, *Nature* (2021)